# Stochastic Online Learning with Feedback Graphs: Finite-Time and Asymptotic Optimality

**Teodor V. Marinov**
Google Research
tvmarinov@google.com

**Mehryar Mohri**
Google Research
& Courant Institute
mohri@google.com

**Julian Zimmert**
Google Research
zimmert@google.com

## Abstract

We revisit the problem of stochastic online learning with feedback graphs, with the goal of devising algorithms that are optimal, up to constants, both asymptotically and in finite time. We show that, surprisingly, the notion of optimal finite-time regret is not a uniquely defined property in this context and that, in general, it is decoupled from the asymptotic rate. We discuss alternative choices and propose a notion of finite-time optimality that we argue is *meaningful*. For that notion, we give an algorithm that admits quasi-optimal regret both in finite-time and asymptotically.

## 1 Introduction

Online learning is a sequential decision making game in which, at each round, the learner selects one arm (or expert) out of a finite set of $K$ arms. In the stochastic setting, each arm admits some reward distribution and the learner receives a reward drawn from the distribution corresponding to the arm selected. In the *bandit setting*, the learner observes only that reward [Lai et al., 1985, Auer et al., 2002a,b], while in the *full information setting*, the rewards of all $K$ arms are observed [Littlestone and Warmuth, 1994, Freund and Schapire, 1997].

Both settings are special instances of a more general model of online learning with side information introduced by Mannor and Shamir [2011], where the information supplied to the learner is specified by a *feedback graph*. In an undirected feedback graph, each vertex represents an arm and an edge between between arm $v$ and $w$ indicates that the reward of $w$ is observed when $v$ is selected and vice-versa. The bandit setting corresponds to a graph reduced to self-loops at each vertex, the full information to a fully connected graph. The problem of online learning with stochastic rewards and feedback graphs has been studied by several publications in the last decade or so. The performance of an algorithm in this problem is expressed in terms of its pseudo-regret, that is the difference between the expected reward achieved by always pulling the best arm and the expected cumulative reward obtained by the algorithm.

The UCB algorithm of Auer et al. [2002a] designed for the bandit setting forms a baseline for this scenario. For general feedback graphs, Caron et al. [2012] designed a UCB-type algorithm, UCB-N, as well as a closely related variant. The pseudo-regret guarantee of UCB-N is expressed in terms of the most favorable *clique covering* of the graph, that is its partitioning into cliques. This guarantee is always at least as favorable as the bandit one [Auer et al., 2002a], which coincides with the specific choice of the trivial clique covering. However, the bound depends on the ratio of the maximum and minimum mean reward gaps within each clique, which, in general, can be quite large.

Cohen et al. [2016] presented an action-elimination-type algorithm [Even-Dar et al., 2006], whose guarantee depends on the least favorable maximal independent set. While there are instances in which this guarantee is worse compared to the bound presented in Caron et al. [2012], in general it

could be much more favorable compared to the clique partition guarantee of Caron et al. [2012]. The algorithm of Cohen et al. [2016] does not require access to the full feedback graph, but only to the out-neighborhood of the arm selected at each round and the results also hold for time-varying graphs. Later, Lykouris et al. [2020] presented an improved analysis of the UCB-N algorithm based on a new layering technique, which showed that UCB-N benefits, in fact, from a more favorable guarantee based on the independence number of the graph, at the price of some logarithmic factors. Their analysis also implied a similar guarantee for a variant of arm-elimination and Thompson sampling, as well as some improvement of the bound of Cohen et al. [2016] in the case of a fixed feedback graph. Buccapatnam et al. [2014] gave an action-elimination-type algorithm [Even-Dar et al., 2006], UCB-LP, that leverages the solution of a linear-programming (LP) problem. The guarantee presented depends only on the domination number of the graph, which can be substantially smaller than the independence number. A follow-up publication [Buccapatnam et al., 2017a] presents an analysis for an extension of the scenario of online learning with stochastic feedback graphs.

We will show that the algorithms just discussed do not achieve asymptotically optimal pseudo-regret guarantees and that it is also unclear how tight their finite-time instance-dependent bounds are. Wu et al. [2015] and Li et al. [2020] proposed asymptotically optimal algorithms with matching lower bounds. However, the corresponding finite-time regret guarantees are far from optimal and include terms that can dominate the pseudo-regret for any reasonable time horizon.

We briefly discuss other work related to online learning with feedback graphs. When rewards are adversarial, there has been a vast amount of work studying different settings for the feedback graph such as the graph evolving throughout the game or the graph not being observable before the start of each round [Alon et al., 2013, 2015, 2017]. The setting in which only noisy feedback is provided by the graph is addressed in Kocák et al. [2016]. First order regret bounds, that is bounds which depend on the reward of the best arm, are derived in Lykouris et al. [2018], Lee et al. [2020]. The setting of sleeping experts is studied in Cortes et al. [2019]. Cortes et al. [2020] study stochastic rewards when the feedback graph evolves throughout the game, however, they do not assume that the rewards and the graph are statistically independent. Another instance in which the feedback and rewards are correlated is that of online learning with abstention [Cortes et al., 2018]. In this setting the player can choose to abstain from making a prediction. The more general problem of Reinforcement Learning with graph feedback has been studied by Dann et al. [2020]. For additional work on online learning with feedback graphs we recommend the survey of Valko [2016].

We revisit the problem of stochastic online learning with feedback graphs, with the goal of devising algorithms that are optimal, up to constants, both asymptotically and in finite time. We show that, surprisingly, the notion of optimal finite-time regret is not a uniquely defined property in this context and that, in general, it is decoupled from the asymptotic rate. Let $T$ denote the time horizon and $\mathsf{Reg}_{\mathcal{A}}(T)$ the pseudo-regret of algorithm $\mathcal{A}$ after $T$ rounds. When $\mathcal{A}$ is clear from the context, we drop the subscript. It is known that $c^*$, the value of the LP considered by Buccapatnam et al. [2014], Wu et al. [2015], Li et al. [2020], is asymptotically a lower bound for $\mathsf{Reg}_{\mathcal{A}}(T)/\log(T)$. We prove that no algorithm $\mathcal{A}$ can achieve a finite-time pseudo-regret guarantee of the form $\mathsf{Reg}_{\mathcal{A}}(T) \leq O(c^*\log(T))$. Moreover, we show that there exists a feedback graph $G$ for which *any* algorithm suffers a regret of at least $\Omega\left(K^{\frac{1}{8}}\left(c^* + \frac{1}{\Delta_{\min}}\right)\right)$, where $\Delta_{\min}$ is the minimum reward gap. We discuss alternative choices and propose a notion of finite-time optimality that we argue is *meaningful*, based on a regret quantity $d^*$ that we show any algorithm must incur in the worst case. For that notion, we give an algorithm whose pseudo-regret is quasi-optimal, both in finite-time and asymptotically and can be upper bounded by $O(c^*\log(T) + d^*)$.

## 2 Learning scenario

We consider the problem of online learning with stochastic rewards and a fixed undirected feedback graph. As in the familiar multi-armed bandit problem, the learner can choose one of $K \geq 1$ arms. Each arm $i \in [K]$ admits a reward distribution, with mean $\mu_i$. For all our lower bounds, we assume that the distribution of the reward of each arm is Gaussian with variance $1/\sqrt{2}$. For our upper bounds, we only assume that the distribution of each arm is sub-Gaussian with variance proxy bounded by 1. We assume that the means are always bounded in $[0, 1]$. For arm $i$, we denote by $\Delta_i = \mu^* - \mu_i$ its mean gap to the best $\mu^* = \max_{i \in [K]} \mu_i$. We will also denote by $\Delta_{\min}$ the smallest and by $\Delta_{\max}$ the largest of these gaps. At each round $t \in [T]$, the learner selects an arm $i_t$ and receives a reward $r_{t,i_t}$ drawn from the reward distribution of arm $i_t$. In addition to observing that reward, the learner

observes the reward of some other arms, as specified by an undirected graph $G = (V, E)$, where the vertex set $V$ coincides with $[K]$: an edge $e \in E$ between vertices $i$ and $j$ indicates that the learner observes the reward of arm $j$ when selecting arm $i$ and vice-versa. We will denote by $N_i$ the set of neighbors of arm $i$ in $G$, $N_i = \{j \in V : (i, j) \in E\}$, and will assume self-loops at every vertex, that is, we have $i \in N_i$ for all $i \in V$. The objective of the learner $\mathcal{A}$ is to minimize its *pseudo-regret*, that is the expected cumulative gap between the reward of an optimal arm $i^*$ and its reward:

$$\mathsf{Reg}(T) = \mathbb{E}\left[\sum_{t=1}^{T}(r_{t,i^*} - r_{t,i_t})\right] = \mu^* T - \mathbb{E}\left[\sum_{t=1}^{T} r_{t,i_t}\right],$$

where the expectation is taken over the random draw of a reward from an arm's distribution and the possibly randomized selection strategy of the learner. In the following, we may sometimes abusively use the shorter term regret instead of pseudo-regret. We will denote by $I^*$ the set of optimal arms, that is, arms with mean reward $\mu^*$, and, for any $t \in [T]$ will denote by $r_t$ the vector of all rewards $r_{t,i}$ at time $t$. When discussing asymptotic or finite-time optimality, we assume the setting of Gaussian rewards.

We will assume an *informed setting* where the graph $G$ is fixed and accessible to the learner before the start of the game. Our analysis makes use of the following standard graph theory notions [Goddard and Henning, 2013]. A subset of the vertices is *independent* if no two vertices in it are adjacent. The *independence number* of $G$, $\alpha(G)$, is the size of the maximum independent set in $G$. A *dominating set* of $G$ is a subset $S \subseteq V$ such that every vertex not in $S$ is adjacent to $S$. The *domination number* of $G$, $\gamma(G)$, is the minimum size of a dominating set. It is known that for any graph $G$, we have $\gamma(G) \le \alpha(G)$. The difference between the domination and independence numbers can be substantial in many cases. For example, for a star graph with $n$ vertices, we have $\gamma(G) = 1$ and $\alpha(G) = n - 1$. In the following, in the absence of any ambiguity, we simply drop the graph arguments and write $\alpha$ or $\gamma$. We will denote by $\mathcal{D}(G')$ the minimum dominating set of a sub-graph $G' \subseteq G$ and by $\mathcal{I}(G')$ the maximum independent set. When the minimum dominating set is not unique, $\mathcal{D}(G')$ can be selected in an arbitrary but fixed way.

## 3 Sub-optimality of previous algorithms

In this section, we discuss in more detail the previous work most closely related to ours [Buccapatnam et al., 2014, Wu et al., 2015, Buccapatnam et al., 2017b, Li et al., 2020] and demonstrate their sub-optimality. A summary of our comparison can be found in Table 1. These algorithms all seek to achieve instance-dependent optimal regret bounds by solving and playing according to the following linear program (LP), which is known to characterize the instance-dependent asymptotic regret for this problem when the rewards follow a Gaussian distribution:

$$c^*(\Delta, G) \coloneqq \min_{x \in \mathbb{R}_+^K} \langle x, \Delta \rangle \qquad s.t. \sum_{j \in N_i} x_j \ge \frac{1}{\Delta_i^2}, \; \forall i \in [K] \smallsetminus I^*. \tag{LP1}$$

We note that these prior works' algorithms can work in more general settings, but we will restrict our discussion to their use in the informed setting with a fixed feedback graph that we consider in this study.

The UCB-LP algorithm of Buccapatnam et al. [2014, 2017b] is based on the following modification of LP1: $\min_{x \in \mathbb{R}_+^K} \langle x, 1 \rangle$ subject to $\sum_{j \in N_i} x_j \ge 1$, for all $i \in [K]$, in which the gap information is eliminated, working with gaps such that $\Delta_{\min} = \Theta(\Delta_{\max})$. This modified problem is the LP relaxation of the minimum dominating set integer program of graph $G$.

The algorithm first solves this minimum dominating set relaxation and then proceeds as an action-elimination algorithm in $O(\log(T))$ phases. During the first $O(\log(K))$ rounds, their algorithm plays by exploring based on the solution of their LP. Once the exploration rounds have concluded, it simply behaves as a bandit action-elimination algorithm. We argue below that this algorithm is sub-optimal, in at least two ways.

**Star graph with equal gaps.** Consider the case where the feedback graph is a *star graph* (Figure 1(a)): there is one root or revealing vertex $r$ adjacent to all other vertices. In our construction, the optimal arm is chosen uniformly at random among the leaves of the graph. The rewards are chosen so that all sub-optimal arms admit the same expected reward with gap to the best $\Delta \le O(1/K^{1+\epsilon})$, $\epsilon > 0$.

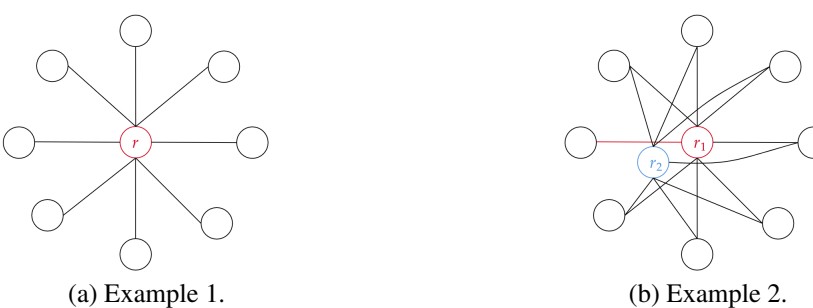

(a) Example 1.        (b) Example 2.

Figure 1: Sub-optimality examples

Table 1: Summary of the regret bounds for Figure 1 (b.)

| | $c^*$ | Min-max rate | Caron et al. (UCB-N), Cohen et al. (Action elimination), Buccapatnam et al. Buccapatnam et al. | Lykouris et al. (UCB-N, Thompson Sampling) | Wu et al. Li et al. | Theorem 5.2 |
|---|---|---|---|---|---|---|
| Regret bound on Example 1 (b.): | $O\left(\frac{\log(T)}{\Delta_{\min}}\right)$ | $O(\sqrt{KT})$ | $O\left(\frac{K\log(T)}{\Delta_{\min}}\right)$ | $O\left(\frac{K\log^2(T)}{\Delta_{\min}}\right)$ | $O\left(\frac{\log(T)}{\Delta_{\min}} + \frac{K}{\Delta_{\min}^2}\right)$ | $O\left(\frac{\log(T)}{\Delta_{\min}}\right)$ |

In this case, an optimal strategy consists of playing the revealing arm for $\Theta(1/\Delta^2)$ rounds to identify the optimal arm, and thus incurs regret at most $O(\frac{1}{\Delta})$. On the other hand the UCB-LP strategy incurs regret at least $\Omega(\frac{K\log(T)}{\Delta})$. Even if we ignore the dependence on the time horizon, the dependence on $K$ is clearly sub-optimal.

**Sub-optimality of using the minimum dominating set relaxation.** In the second problem instance, given in Figure 1(b), we consider a star-like graph in which we have a revealing vertex $r_1$, adjacent to all other vertices. We also have an "almost" revealing vertex $r_2$ which is adjacent to all vertices but a single leaf vertex (leaves are the vertices with degree 1 and 2 in this case). The optimal arm is again chosen uniformly among the leaves. Rewards are set so that the gap at $r_1$ is $\Delta_{\max}$ and the remaining gaps are $\Delta_{\min}$. The solution to the LP of Buccapatnam et al. [2014, 2017b] puts all the weights on $r_1$. However, the optimal policy for this problem consists of playing $r_2$ and the leaf vertex not adjacent to $r_2$ until all arms but the optimal arm are eliminated. The instance optimal regret in this case is $O(\frac{\log(T)}{\Delta_{\min}})$, while UCB-LP incurs regret $\Omega(\frac{K\log(T)}{\Delta_{\min}})$.

Next, we discuss [Wu et al., 2015] and [Li et al., 2020]. Their instance-dependent algorithms are based on iteratively solving empirical approximations to LP1. For simplicity, we only discuss the instance-dependent regret bound in [Li et al., 2020]. A similar bound can be found in [Wu et al., 2015]. Let $x(\Delta)$ denote the solution of LP1 and define the following perturbed solution

$$x_i(\Delta, \epsilon) = \sup\{x_i(\Delta') : |\Delta'_i - \Delta_i| \le \epsilon, \forall i \in [K]\}.$$

The solution $x(\Delta, \epsilon)$ is the solution of LP1 with $\epsilon$-perturbed gaps. [Li et al., 2020][Theorem 4] states that the expected regret of their algorithm is bounded as follows: $\text{Reg}(T) \le O\left(\sum_{i\in[K]} \log(T) x_i(\Delta, \epsilon)\Delta_i + \sum_{t=1}^T \exp\left(-\frac{\beta(t)\epsilon^2}{K}\right) + K\right)$, for any $\epsilon > 0$ and $\beta(t) = o(t)$. For the standard bandit problem with Gaussian rewards, we can compute the perturbed solution: $x_i(\Delta, \epsilon) = \max\left(\frac{1}{(\Delta_i+\epsilon)^2}, \frac{1}{(\Delta_i-\epsilon)^2}\right)$. Thus, for a meaningful regret bound, we would need $\epsilon = \Theta(\Delta_{\min})$. If $\epsilon$ is much smaller, then the term $\sum_{t=1}^T \exp\left(-\frac{\beta(t)\epsilon^2}{K}\right)$ becomes too large and otherwise we risk making $x_i(\Delta, \epsilon)$ too large. To analyze the second term more carefully, we allow $\beta(t) = t$. The first $\frac{K}{\Delta_{\min}^2}$ terms of $\sum_{t=1}^T \exp\left(-\frac{\beta(t)\epsilon^2}{K}\right)$ are now at least $\frac{1}{e}$ and thus this sum is at least $\sum_{t=1}^T \exp\left(-\frac{\beta(t)\epsilon^2}{K}\right) \ge \frac{K}{e\Delta_{\min}^2}$. Thus, the bandit regret bound evaluates to at least $\Omega\left(\sum_{i\in[K]} \frac{\log(T)}{\Delta_i} + \frac{K}{\Delta_{\min}^2}\right)$. While this bound is asymptotically optimal, since the second term does not have dependence on $T$, it admits a very poor dependence on the smallest gap. We can repeat the argument above with a star-graph construction in which the revealing vertex has gap $\Delta_{\min}$. In this case, the optimal strategy given by the solution to LP1 consists of playing the revealing vertex for $\frac{1}{\Delta_{\min}^2}$ times and incurs regret at most $O\left(\frac{\log(T)}{\Delta_{\min}}\right)$. The regret bound of the algorithm of Li et al. [2020], however, amounts to $\Omega\left(\frac{\log(T)}{\Delta_{\min}} + \frac{K}{\Delta_{\min}^2}\right)$.

# 4   Instance-dependent finite-time bounds

In this section, we provide an in-depth discussion of what finite-time optimality actually means. Finite-time bounds are statements of the form $\mathsf{Reg}(T) \leq f(T)$, which hold for any $T > 0$. Specifically, we are considering functions of the type $f(T) = c^* \log(T) + d$,[1] which we know to exist from prior work.[2] The question of what the optimal expression of $d$ might be seems easy to answer at first. Indeed, for bandits with Gaussian rewards, one can achieve $d = O(\sum_i \Delta_i)$, which in general is much smaller than $c^* = \Omega(\sum_i \frac{1}{\Delta_i})$ [Lattimore and Szepesvári, 2020], and hence will be dominated by the time-dependent part of the regret for almost all reasonable lengths of the time horizon $T$. In full information, we obtain a meaningful optimal value $d^*$ for a given gap vector by considering the worst-case regret of any algorithm under any permutation of the arms. This leads to $d^* = O\left(\frac{\ln(K)}{\Delta_{\min}}\right)$ [Mourtada and Gaïffas, 2019]. Note that in the full information setting we have $c^* = 0$.

One might hope for a similar structure for feedback graphs, where the optimal $d$ depends only on the "full-information structure", that is the gaps of arms neighboring an optimal arm. All other arms contribute to $c^*$ and we might assume that their complexity is already captured in the $c^* \log(T)$ term as it is the case for bandits. However, the situation is more complicated, as we show next.

**Theorem 4.1.** *For any $K \geq 32$ and $\Delta_{\min} = \mathcal{O}(\frac{1}{\sqrt{K}})$, there exists a graph $G$ with $K$ vertices, such that for any algorithm there exists an instance with unique best arm and minimal gap $\Delta_{\min}$ for which for any $T \geq \Omega(\frac{K^{3/4}}{\Delta_{\min}})$, the regret is at least $\mathsf{Reg}(T) = \Omega\left(K^{\frac{1}{8}}\left(c^* + \frac{1}{\Delta_{\min}}\right)\right)$.*

Theorem 4.1 shows that there exists a problem instance, in which $d \gg c^*$ dominates the finite time regret for any $T \leq O(\exp(K^{1/8}))$. We note that this is not simply due to the full-information structure of the feedback graph $G$, as $c^*$ is positive. Furthermore, combined with examples shown in Section 7, Theorem 4.1 suggests that there is no simple characterization of $d$ independent of the graph in terms of $c^*$, e.g., $d = \Theta(c^*/\Delta_{\min})$. While $d = \Theta(K^{1/8}c^*)$ holds for the example instance in Theorem 4.1, there exists a non-trivial family of graphs, for which for any rewards instance, we have $d = O(c^*)$, so the shape and relevance of $d$ is highly graph-dependent.

Having established that $d$ could be the dominating term in the regret for any reasonable time horizon, we now discuss the hardness of defining an optimal $d$. Let us first consider a simple two-arm full-information problem and inspect algorithms of the style: "Play arm 1, unless the cumulative reward of arm 2 exceeds that of arm 1 by a threshold of $\tau$." This kind of algorithm has small regret (small $d$) if arm 1 is optimal, and large otherwise. Tuning $\tau$ yields different trade-offs between the two scenarios. The same issue appears in learning with graph feedback on a larger scale. Given two instances defined by gap vectors $\Delta$ and $\Delta'$ respectively, an agent can trade off the constant regret part $d(\Delta)$ and $d(\Delta')$ in the two instances. Take for example two algorithms $\mathcal{A}$ and $\mathcal{B}$ and assume the respective values of $c^*$ and $d$ for the two instances and algorithms are given by Table 2. As we show in Appendix A, there exist a feedback graph and instances that are consistent with the table. Which algorithm is more "optimal", $\mathcal{A}$ or $\mathcal{B}$? $\mathcal{B}$ ensures that $\max_{\delta \in \{\Delta, \Delta'\}} \frac{d(\delta)}{c^*} = \mathcal{O}(1)$ and we can

|  | $c^*$ | $d$ for Alg. $\mathcal{A}$ | $d$ for Alg. $\mathcal{B}$ |
|---|---|---|---|
| Instance $\Delta$ | $C/3$ | $C/3$ | $4C/9$ |
| Instance $\Delta'$ | $4C\varepsilon$ | $C/2$ | $4C\varepsilon$ |

Table 2: Comparison of $c^*$ and $d$.

write the regret function as $f(T) = \mathcal{O}(c^* \log(T))$ without the need of a constant term $d$ at all. This algorithm minimizes the competitive ratio of $d$ and $c^*$. $\mathcal{A}$ minimizes the worst-case absolute regret $\mathcal{A} = \operatorname{argmin}_{a \in \{\mathcal{A}, \mathcal{B}\}} \max_{\delta \in \{\Delta, \Delta'\}} d(\delta, a)$.

Thus, we argue that the notion of optimality is subject to a choice and that there is no unique correct answer. In this paper, we opt for the worst-case absolute regret, minimized by $\mathcal{A}$ in the above example, for the following reasons: 1. Theorem 4.1 shows that a constant competitive ratio is generally unachievable; 2. Optimizing regret in general is a different objective than that of competitive

---

[1]When the problem parameters are clear from the context, we will write $c^*$ instead of $c^*(\Delta, G)$.

[2]While obtaining exact asymptotic optimality $\lim_{T \to \infty} \frac{f(T)}{\log(T)} = c^*$ would be ideal, we settle for optimality up to a multiplicative constant in our upper bounds.

ratio. Optimizing for a mixture implies a counter-intuitive preference such as: "In a hard environment where I cannot avoid suffering a loss of 1000, it does not matter much if I suffer an additional 1000 on top, as long as I do better on easier environments." 3. Moreover, note that, even if one were interested in optimizing the competitive ratio between $c^*$ and $d$, it is unclear if one could achieve that objective computationally efficiently.

We present our final definition for an optimal notion of $d^*$ in Section 6. The high level idea is to take all confusing instances, where the means are perturbed by less than $\Delta_s$ and consider the worst-case regret any algorithm suffers over these instances until identifying all gaps up to $\Delta_s$ precision.

## 5 Algorithm and regret upper bounds

Our algorithm works by approximating the gaps $(\Delta_i)_{i \in [K]}$ and then solving a version of LP1. First, note that all arms $i$ with gaps $\Delta_i \leq \frac{1}{T}$ can be ignored as the total contribution to the regret is at most $O(1)$. We now segment the interval $[\frac{1}{T}, 1]$, containing each relevant gap, into sub-intervals $[2^{-s}, 2^{-s+1}]$, where $s \in [[\log_2(T)]]$. The algorithm now proceeds in phases corresponding to each of the $\lceil \log_2(T) \rceil$ intervals. During phase $s$, all arms with gaps $\Delta_i \in [2^{-s}, 2^{-s+1}]$ will be observed sufficiently many times to be identified as sub-optimal.

For phase $s$, let $\Delta_s = 2^{-s}$ denote the smallest possible gap that can be part of the interval $[2^{-s}, 2^{-s+1}]$ and define the clipped gap vector as $\Delta^s \in \mathbb{R}^K, \Delta_i^s = \Delta_s \vee \Delta_i$. Further, define the set $\Gamma_s = \{i \in [K] : \Delta_i \leq 2\Delta_s\}$. $\Gamma_s$ consists of all optimal arms $I^*$ and all sub-optimal arms with gaps small enough, making them impossible to distinguish from optimal arms. Define the following LP

$$\min_{x \in \mathbb{R}^{[K]}} \langle \Delta^s, x \rangle \qquad s.t. \sum_{j \in N_i} x_j \geq \frac{1}{\Delta_s^2}, \ \forall i \in \Gamma_s. \tag{LP2}$$

For any arm $i$ such that $\Delta_i \in [2^{-s}, 2^{-s+1}]$ observing $i$ for $\frac{1}{\Delta_s^2}$ times is sufficient to identify $i$ as a sub-optimal arm. Further, information theory dictates that $i$ needs to be observed at least $\frac{1}{\Delta_{s-1}^2}$ times to be distinguished as sub-optimal. Thus, the constraints of LP2 are necessary and sufficient for identifying the sub-optimal arms $i$ with $\Delta_i \in [2^{-s}, 2^{-s+1}]$. Furthermore, since there is no sufficient information to distinguish between any two arms $i$ and $j$ with gaps $\Delta_i \leq \Delta_j < \Delta_s$, we choose to treat all of them as equal in the objective of the LP. Indeed, Lemma 6.1 shows that for any graph $G$ and any algorithm, there exists an assignment of the gaps $\Delta_i < \Delta_s$ so that the algorithm will suffer regret proportional to the value of LP2.

In practice, it is impossible to devise an algorithm that solves and plays according to LP2 because even during phase $s$, there is still no complete knowledge of the gaps $\Delta_i > \Delta_s$, but, rather only empirical estimators, and so there is no access to $\Delta^s$. We also replace the constraints by a confidence interval term of the order $\frac{\log(1/\delta_s)}{\Delta_s^2}$. This enables us to bound the probability of failure for the algorithm by $\delta_s$ during phase $s$. We note that standard choices of $\delta_s$ such as $\delta_s = \Theta\left(\frac{1}{T}\right)$ from UCB-type strategies will result in a regret bound that has a sub-optimal time-horizon dependence. This suggests that a more careful choice of $\delta_s$ must be determined.

### 5.1 Algorithm

To describe our algorithm, we will adopt the following definitions and notation. Let $\tau_s$ denote the last time-step of phase $s$. We will denote by $n_i(s)$ the total number of times the reward of arm $i$ is observed up to and including $s$, $n_i(s) = \sum_{t=1}^{\tau_s} \mathbb{I}(i_t \in N_i)$, and by $r_i(s)$ the average reward observed, $\hat{r}_i(s) = [\sum_{t=1}^{\tau_s} r_{t,i} \mathbb{I}(i_t \in N_i)]/n_i(s)$. We also denote by $\hat{\Delta}_i(s)$ a lower bound on $\Delta^s$ with a shrinking confidence interval $b_i(s)$ and by $\hat{\Gamma}_s$ the empirical version of the set $\Gamma_s$:

$$\hat{\Delta}_i(s) = \Delta_s \vee \max_{j \in [K]} \hat{r}_j(s) - b_j(s) - \hat{r}_i(s) - b_i(s), \text{ where } b_i(s) = \sqrt{\frac{3\alpha \log(\frac{K}{\Delta_{s+1}})}{n_i(s)}}$$

$$\hat{\Gamma}_s := \{i \in [K] \mid \hat{\Delta}_i(s-1) \leq 2\Delta_s\}$$

---
**Algorithm 1:** Algorithm based on LP3
---
**Input :** Graph $G = (V, E)$, confidence parameter $\delta$, time horizon $T$

1 **Initialize** $t = 0$, $s = 0$, $\hat{r}_i(0) = 0$, $\forall i \in [K]$
2 Compute (approximate) minimum dominating set $\hat{\mathcal{D}}(G)$
3 **while** $s \leq \lceil \log(K) \rceil$ **do**
4     Play each arm $i \in \hat{\mathcal{D}}(G)$ for $\frac{\alpha' \log(\frac{K}{\Delta_{s+1}})}{\Delta_s^2}$ rounds
5     Update $t$ and $s$.
6 **while** $t \leq T$ **do**
7     Compute a (approximate) solution $x^*_{LP3}$ to LP3.
8     Play each action $i$ for $\lceil (x^*_{LP3})_i \rceil$ rounds and update $t$.
9     Update the phase $s{+} = 1$.
---

Our algorithm solves an empirical version of (LP2) at each phase, which is the following LP:

$$\min_{x \in \mathbb{R}_+^K} \langle x, \hat{\Delta}(s-1) \rangle \qquad s.t. \sum_{j \in N_i} x_j \geq \frac{\alpha' \log(\frac{K}{\Delta_{s+1}})}{\Delta_s^2}, \forall i \in \hat{\Gamma}_s,$$

$$\sum_{j \in N_i} x_j \geq \frac{\alpha'}{\hat{\Delta}_i^2(s-1)}, \forall i \notin \hat{\Gamma}_s . \qquad \text{(LP3)}$$

Pseudocode can be found in Algorithm 1. In the first $\lceil \log(K) \rceil$ rounds, the algorithm just plays according to the minimum dominating set of $G$. This is because there is not enough information regarding any of the gaps. Denote the approximate solution of LP3 as $x^*_{LP3}$ at phase $s$. Then at every round of phase $s$ we play each arm exactly $\lceil (x^*_{LP3})_i \rceil$ many times. Phase $s$ then ends after $\sum_{j \in [K]} \lceil (x^*_{LP3})_i \rceil$ rounds. We note that it is sufficient to approximately solve LP3 so that the constraints are satisfied up to some multiplicative factor and the value of the solution is bounded by a multiplicative factor in the value of the LP.

### 5.2 Regret bound

The first step in the regret analysis of Algorithm 1 is to relate the value of LP3 to the value of LP4 based on the true gaps given below.

$$\min_{x \in \mathbb{R}_+^K} \langle x, \Delta^s \rangle \qquad s.t. \sum_{j \in N_i} x_j \geq \frac{\alpha' \log(\frac{K}{\Delta_{s+1}})}{\Delta_s^2}, \forall i \in \Gamma_s,$$

$$\sum_{j \in N_i} x_j \geq \frac{\alpha'}{\Delta_i^2}, \forall i \notin \Gamma_s. \qquad \text{(LP4)}$$

We do so by showing that $\hat{\Gamma}_{s+1} \subseteq \Gamma_s$ and that $\hat{\Delta}(s) = \Theta(\Delta^s)$. This allows us to upper upper bound the value of LP3 by the value of LP4 in the following way.

**Lemma 5.1.** *Let $D_{LP3}(s)$ be the value of LP3 at phase $s$ and let $D_{LP4}(s)$ be the value of LP4 at phase $s$. For any $s \geq \log(K) \vee 10$ holds that $D_{LP3}(s+1) \leq 4D_{LP4}(s)$, with probability at least $1 - 3\left(\frac{\Delta_{s/2+1}}{K}\right)^{\alpha-2}$. Further, for any $s \geq \log(|I^*|/(4\Delta_{\min})) \vee 10$ it holds that the regret incurred for playing according to LP3 is at most $16\alpha' c^*(G, \mu)$ with the same probability.*

Lemma 5.1 shows that playing Algorithm 1 is already asymptotically optimal, as the incurred regret during any phase $s \geq \log(|I^*|/\Delta_{\min})$ starts being bounded by $O(c^*)$. There are two challenging parts in proving Lemma 5.1. First is how to handle the concentration of $\hat{\Delta}_i(s)$ for actions $i \notin \hat{\Gamma}_s$ which have been eliminated prior to phase $s$. This challenge arises because $\alpha'$ needs to be set as a time-independent parameter as the time-horizon part of the regret incurred by the algorithm will depend on $\alpha'$. We notice that for any phase $s \geq 2$ the event that the empirical reward, $\hat{r}_i(t)$, concentrates uniformly around its mean $\mu_i$ in the interval $t \in [s/2, s]$ can be controlled with high probability. This in turn guarantees that the empirical gap estimator $\hat{\Delta}(t)$ is small enough and hence action $i$ is observed sufficiently many times in phases $[s/2, s]$.

The second challenge is to analyze the regret of the solution of LP3 directly, for any $s \geq \log\left(\frac{|I^*|}{K}\right)$ so that we can bound this regret by $c^*$. The key observation is that there exists a $\hat{x}^*$ which is feasible (with high probability) for LP1 with the property that $\langle \hat{x}^*, \hat{\Delta}(s) \rangle \leq O(c^*)$ and further $\sum_{i \notin I^*} x^*_{LP3,i} \hat{\Delta}_i(s) \leq 2 \sum_{i \notin I^*} \hat{x}^*_i \hat{\Delta}_i(s)$. This is sufficient to conclude that $D_{LP3}(s) \leq O(c^*)$

Lemma 5.1 can now be combined with the observation that the constraints of LP2 are a subset of the constraints of LP4, up to a logarithmic factor in $\frac{1}{\Delta_{\min}}$, to argue the following upper regret bound.

**Theorem 5.2.** *Let* $d^*(G, \mu) = \max_{s \leq \log(|I^*|/\Delta_{\min})} D_{LP2}(s)$. *There exists an algorithm with expected regret* $\mathsf{Reg}(T)$ *bounded as*

$$\mathsf{Reg}(T) \leq O\left(\log^2\left(\frac{1}{\Delta_{\min}}\right) d^* + \log(T) c^* + \gamma(G) K \log(K)\right) \wedge \tilde{O}(\sqrt{\alpha(G)T}).$$

We note that Algorithm 1 can incur additional regret of order $O(K)$ per phase due to the rounding, $\lceil x^*_{LP3} \rceil$, of the solution to LP3. Thus its regret will only be asymptotically optimal in the setting when $\Delta_{\min} \leq O(1/K)$. To fix this minor issue, we present an algorithm with more careful rounding in Appendix B.1, which enjoys the regret bound of Theorem 5.2.

## 6 Regret lower bounds

**Lower bound with $d^*$.** We are able to show the following result for any algorithm.

**Lemma 6.1.** *Fix any instance* $\mu$ *s.t.* $\mu_i \leq 1 - 2\Delta_s, i \in I^*$. *Let* $\Lambda_s(\mu)$ *be the set of problem instances with means* $\mu' \in \mu + [0, 2\Delta^s]^k$. *Then for any algorithm, there exists an instance in* $\Lambda_s(\mu)$ *such that the regret is lower bounded by* LP2.

Motivated by Lemma 6.1, the quantity $d^*(G, \mu)$ is a meaningful definition of finite-time optimality. We note that $d^*$ is indeed independent of the time-horizon and only depends on the topology of $G$ and the instance $\mu$. The result in Lemma 6.1 is a companion to the upper bound in Theorem 5.2. It shows that for any instance $\mu$ and number of observations which are not sufficient to distinguish the arms with smallest positive gaps as sub-optimal, any algorithm will necessarily incur large regret of order $d^*$. This happens because the algorithm will not be able to distinguish $\mu$ from some environment $\mu'$ which is identical to $\mu$ except for the reward of a single arm which is only slightly perturbed.

The definition of $d^*$ as a maximum over different values of $s$ might seem surprising, as one could expect that the value of LP2 strictly increases when $s$ grows, after all this is precisely what happens in the bandit setting. This is not the case for general graphs, where the value can also decrease between phases $s$ and $s + 1$. Intuitively this happens when the approximate minimum weighted dominating set chosen by the LP's solution increases between phases.

The result in Lemma 6.1 has a min-max flavor in the sense that all possible instances which are close to $\mu$ are considered. It is reasonable to ask if $d^*$ can be further bounded by a favorable instance-dependent quantity. The answer to this question is complicated and certainly depends on the topology of the feedback graph as we show next.

**Sketch of proof of Theorem 4.1.** We now show that any finite time term $d$ has to exceed $c^*$ by at least a multiplicative polynomial factor in the number of actions $K$. To do so we exhibit a specific feedback graph $G$, found in Figure 2, on which any algorithm will have to incur regret at least $\Omega(K^{\frac{1}{8}} c^*)$ for some $\mu$ s.t. $c^* \geq \frac{1}{\Delta_{\min}}$.

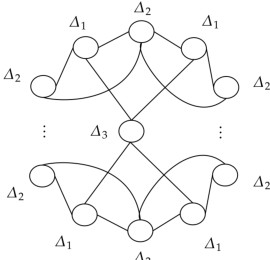

Figure 2: Reinforced wheel.

Formally the graph is defined to have a vertex set $V = \mathcal{N}_1 \cup \mathcal{N}_2 \cup \mathcal{N}_3$ of $2K + 1$ arms, with each of $\mathcal{N}_i$'s disjoint and $\mathcal{N}_1 = \{2i : 1 \leq i \leq K\}$, $\mathcal{N}_2 = \{2i + 1 : 0 \leq i \leq K\}$, $\mathcal{N}_3 = \{0\}$. The set of edges is defined as follows. Every vertex in $\mathcal{N}_1$ is adjacent to the vertex in $\mathcal{N}_3$ and the $2i$ vertex is adjacent to both $2i + 1$ and $2i - 1$ in $\mathcal{N}_2$ modulo $2K + 1$. Finally vertex $2i + 1$ in $\mathcal{N}_2$ is further adjacent to to the next $\lceil K^{1/8} \rceil$ vertices in $\mathcal{N}_2$ modulo $2K + 1$. The base instance, $\mathcal{E}$, is defined by a scalar $\nu \in [0, 1]$ and gap parameter $\Delta$ so that the expected reward of every action in $\mathcal{N}_1$ is equal to $\nu - \Delta$, the expected reward of every action in $\mathcal{N}_2$ is equal to $\nu - K^{1/4}\Delta$ and the expected reward of the action in $\mathcal{N}_3$

is $\nu - \sqrt{K}\Delta$. We assume that all rewards follow a Gaussian with variance $\frac{1}{\sqrt{2}}$. We denote by $\Delta_1 = \Delta, \Delta_2 = K^{\frac{1}{4}}\Delta, \Delta_3 = \sqrt{K}\Delta$.

The lower bound now fixes an algorithm $\mathcal{A}$ and considers two cases. First, $\mathcal{A}$ could commit to often playing arms in $\mathcal{N}_1$. In this case we show that there could be a large gap to the arms in $\mathcal{N}_1$ which would not be detectable by $\mathcal{A}$ as arms in $\mathcal{N}_2$ are not observed often enough. This is indeed the case as $\mathcal{A}$ needs to play $\Omega(K)$ actions in $\mathcal{N}_1$ to cover $\mathcal{N}_2$. This first case corresponds to assuming that the number of arms played from $\mathcal{N}_2$ is at most $O(K^{\frac{7}{8}}/\Delta_2^2)$ in the first $O(K/\Delta_2^2)$ rounds. The second case considers the scenario in which actions in $\mathcal{N}_2$ are played for more than $\Omega(K^{\frac{7}{8}}/\Delta_2^2)$ times in the first $O(K/\Delta_2^2)$ rounds. In this case, $\mathcal{A}$ would suffer large regret if the gap at actions in $\mathcal{N}_1$ is small enough, so that the optimal strategy is to cover $\mathcal{N}_2$ by playing arms in $\mathcal{N}_1$.

More formally, we begin by showing that there always exists an arm $n^* \in \mathcal{N}_2$ which is observed for only $O(1/\Delta_2^2)$ times. Next, we change the expected reward of $n^*$ depending on which of the above two cases occur. In the first case we change the environment by setting the reward of $n^*$ to have expectation $\nu + \Delta_2$. We can now argue that the regret of $\mathcal{A}$ will be at least $\Omega(K^{\frac{3}{4}}/\Delta)$ as $n^*$ will not be played often enough in the new environment. The value of $c^*$, however, is at most $O(K^{\frac{7}{8}}/\Delta_2) = O(K^{\frac{5}{8}}/\Delta)$, as playing each action in a minimum dominating set over $\mathcal{N}_2$ for $\frac{1}{\Delta_2^2}$ rounds is feasible for LP1. For the second case, we set the expected reward of $n^*$ to equal $\nu$. The optimal strategy now has regret at most $O(\sqrt{K}/\Delta)$ by playing the action in $\mathcal{N}_3$ for $\frac{1}{\Delta^2}$ and every action in $\mathcal{N}_1$ for $\frac{1}{\Delta_2^2}$ rounds. On the other hand $\mathcal{A}$ will incur at least $\Omega(K^{\frac{5}{8}}/\Delta)$, as again $n^*$ is not played often enough. This argument implies the result presented in Theorem 4.1.

# 7 Characterizing the value of $d^*$

Theorem 4.1 suggests that we take into account the topology of $G$ explicitly when trying to bound $d^*$, independently of the instance $\mu$. In this section, we first show a bound on $d^*$ that depends only on independent sets of $G$. Then, we show a set of graphs $G$ for which $d^* \leq O(c^*)$ on any instance $\mu$.

Let us recall the regret bounds presented in [Lykouris et al., 2020]. Denote by $\mathcal{I}(G)$ the set of all independent sets for the graph $G$. Then the regret bounds presented in [Lykouris et al., 2020] are of the order $\mathrm{Reg}(T) \leq O\left(\max_{I \in \mathcal{I}(G)} \sum_{i \in I} \frac{\log^2(T)}{\Delta_i}\right)$. It is possible to show, as we do in Appendix D.1, that $d^*(G, \mu) \leq \max_{I \in \mathcal{I}(G)} \sum_{i \in I} \frac{1}{\Delta_i}$. Thus, our algorithm enjoys regret bounds which are better than what is known for the algorithms studied in [Cohen et al., 2016, Lykouris et al., 2020]. The above bound, however, could be very loose as was discussed in the beginning of the paper, especially when considering star-graphs, as the bound would just reduce to the bandit case. It turns out, however, that $d^* \leq O(c^*)$ in this case. In fact we can state a sufficient condition on $G$ so that $d^* \leq c^* + \frac{|I^*|}{\Delta_{\min}}$ for a more general family of graphs. We begin by defining the following operation on $G$.

**Definition 7.1.** *Let $\sim$ be the equivalence class defined by $u \sim v$ iff $N_u = N_v$ and let $\mathcal{C}$ be the mapping which sends $G$ to the quotient $G/_{\sim}$ through the operation of collapsing any sub-graph of $G$ into its equivalence class.*

We note that $\mathcal{C}$ is well-defined as the relation $\sim$ is an equivalence relation. The equivalence classes defined by $\sim$ are cliques with the following property. For any $v$ in an equivalence class $[v]$ it holds that $u \in [v], \forall u \in N_v$, that is the vertices in the equivalence class clique only have neighbors in the clique to which they belong. An example can be found in Figure 3, where the graph contains four cliques adjacent to the vertex $r$. In general, the vertex $r$ can also be a clique as well and the graph can have multiple disjoint components of the type presented in the figure. For any instance $\mu$ of the problem, this allows us to collapse each equivalence class $[v]$ to a vertex $v$ with the maximum expected reward in $[v]$. The next lemma states a sufficient condition on $G$ under which $d^*$ is bounded.

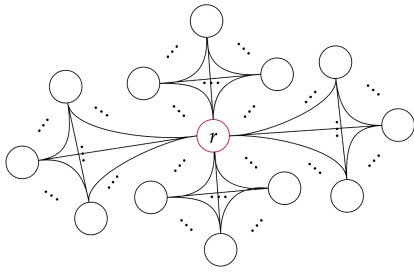

Figure 3: Generalized star-graph.

**Lemma 7.2.** *If the graph $G$ is such that $\mathcal{C}(G)$ has no path of length greater than two between any two vertices, then, for any instance $\mu$, the following inequality holds: $c^* + \frac{|I^*|}{\Delta_{\min}} \geq d^*$.*

# 8  Conclusion

We presented a detailed study of the problem of stochastic online learning with feedback graphs in a finite time setting. We pointed out the surprising issue of defining optimal finite-time regret for this problem. We gave an instance on which no algorithm can hope to match, in finite time, the quantity $c^*$, which characterizes asymptotic optimality. Next, we derived an asymptotically optimal algorithm that is also min-max optimal in a finite-time sense and admits more favorable regret guarantees than those given in prior work. Finally, we described a family of feedback graphs for which matching the asymptotically optimal rate is possible in finite time.

There are several interesting questions that follow from this work. First, while the condition on $\mathcal{C}(G)$ in Lemma 7.2 is sufficient, it is not necessary. For example, a star-like graph in which two leaf vertices are also neighbors will have the property that $d^* \leq O(c^*)$ for any instance $\mu$. We ask what would be a necessary and sufficient condition on $G$ for which $d^* = \Theta(c^*)$ on any instance $\mu$? Another interesting question is how to address the setting of evolving feedback graphs. It is unclear what conditions on the graph sequence would allow us to recover bounds that improve on the existing independence number results. Further, can we use our approach to show improved results for the setting of dependent rewards and feedback graphs studied in [Cortes et al., 2020]? Finally, our methodology crucially relies on the informed setting assumption. We ask if it is possible to achieve similar bounds to Theorem 5.2 in the uninformed setting.

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
