# Contents of Appendix

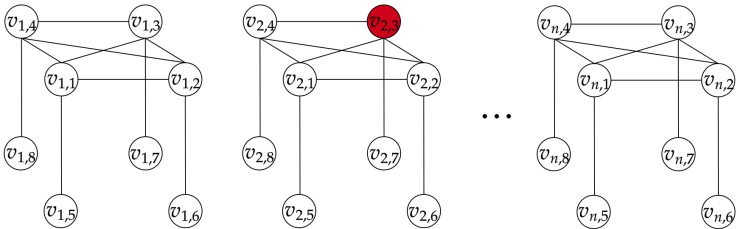

Figure 4: Arm in red is optimal

# A Refined example for hardness of determining optimal $d$

To understand better why it is difficult to define a notion of *optimality* for the constant term $d$ in the finite-time bound, consider the following toy problem. The graph is given by Figure 4. There are $n$ disjoint copies of an open cube graph with 8 vertices each. We let $V_1 = \{\nu_{i,1}, \nu_{i,2}, \nu_{i,3}, \nu_{i,4}\}_{i \in [n]}$ and $V_2 = \{\nu_{i,5}, \nu_{i,6}, \nu_{i,7}, \nu_{i,8}\}_{i \in [n]}$. We assume that we have oracle knowledge of the mean rewards of all arms $\mu(\nu) = \frac{1}{2}$ for any $\nu \in V_2$ and $\mu(\nu) = \frac{1}{2} - \Delta$ for all $\nu \in V_1$, with one exception. There is one arm in $V_1$, chosen uniformly at random, that is optimal with a mean $\mu^* \in \{\frac{1}{2} + 2\Delta, \frac{1}{2} + \varepsilon\Delta\}$. We note that we do not know the index of the optimal arm and so the problem reduces to identifying the optimal arm and the respective environment (i.e. value of $\mu^*$). The best we can do is to collect equally many samples for each arm in $V_1$ until we have sufficient statistics to figure out either the environment or the optimal arm. Under Env. A we need to collect $1/(3\Delta)^2$ samples and under Env. B we need to collect $1/((1 + \epsilon)\Delta)^2$ samples. There are two canonical base strategies corresponding to algorithm $\mathcal{A}$ and $\mathcal{B}$ in Section 4: either play all arms in $V_2$ for $N(env)$ times (Algorithm $\mathcal{A}$), depending on the environment, or play all arms in $V_1$ for $N(env)/4$ many times (Algorithm $\mathcal{B}$). The following table shows the regret each strategy suffers for collecting sufficient samples to distinguish the environments.

|  | Env. A ($\mu^* = \frac{1}{2} + 2\Delta$) | Env. B ($\mu^* = \frac{1}{2} + \varepsilon\Delta$) |
|---|---|---|
| $\mathcal{A}$ (Play $V_1$) | $n/(3\Delta)$ | $n/((1 + \varepsilon)\Delta)$ |
| $\mathcal{B}$ (Play $V_2$) | $4n/(9\Delta)$ | $4n\varepsilon/((1 + \varepsilon)^2\Delta)$ |

Under Env. A we have $c^*_{\text{Env. A}} = \frac{n}{(3\Delta)}$ and under Env. B we have $c^*_{\text{Env. B}} = \frac{n\varepsilon}{(1+\epsilon)^2\Delta}$. Which strategy is the "optimal" one? One possible answer is to say that $\mathcal{A}$ is optimal, since it minimizes the worst-case regret. One might be tempted to say that $\mathcal{B}$ is better, since we can absorb the constant term in the leading $\mathcal{O}(c^* \log(T))$ without the need of adding a constant $d$ at all! That is, $\mathcal{B}$ minimizes the competitive ration.

The implicit assumption made for the second choice of optimality is: "In a bad environment, where it is inevitable to suffer a loss of 100000, suffering an additional 100000 is just as bad as suffering an additional loss of 10 in an environment where one cannot avoid a loss of 10." We argue that this notion of optimality is not aligned with the principle of regret as a benchmark. In regret, unlike the competitive ratio, we care about the absolute value of suboptimality. Hence, we claim that considering strategy 1 optimal in our toy experiment independent of the value of $c^*$ in environment A and B is a meaningful choice. The same argument implies that hiding arbitrarily large constants in the $\mathcal{O}$-notation will obscure critical information about the practicalities of an algorithm, which our work unfortunately does as well. The regret upper bounds presented in this work hide only universal constants which are independent of the problem parameters, including the topology of the feedback graph.

# B Regret upper bound proofs

For the rest of the appendix we are going to assume that each gap $\Delta_i$ is such that $\Delta_i = 2^{-f(i)}$ for some function $f: [K] \to \lceil \log(T) \rceil$. This is without loss of generality as every $\Delta_i$ is in $[2^{-s}, 2^{-s+1}]$ for some $s$. Thus, we can clip every $\Delta_i$ to $2^{-s_i}$ for some $s_i$ and change the constraints and objective of LP1 by at most a factor of 2. Thus the value of $c^*$ would change by at most a factor of 2.

## B.1 Algorithm modification

Since Algorithm 1 plays $\lceil x^*_{LP3,i} \rceil$ we need to take care of the difference $\lceil x^*_{LP3,i} \rceil - x^*_{LP3,i}$. At worst, playing according to the rounded solution of LP3 can result in a $\Omega(K)$ additive factor on top of $D_{LP3}(s)$. This can accumulate regret up to an $\Omega(K\log(T))$ factor in the final bound. Our goal is to give asymptotically optimal bounds together with the finite time bounds and such a term might be sub-optimal in the case when $\Delta_{\min} \geq \omega(\frac{1}{K})$.

To avoid the additional $K$-factor we modify Algorithm 1 in the following way. Note that for any

---

**Algorithm 2:** Modification of Algorithm 1

**input :** Graph $G = (V, E)$, confidence parameter $\delta$, time horizon $T$

1 **Initialize** $t = 0$, $s = 0$, $\hat{r}_i(0) = 0$, $\forall i \in [K]$, $B = [0]^K$

2 Compute (approximate) minimum dominating set $\hat{\mathcal{D}}(G)$

3 **while** $s \leq \lceil \log(K) \rceil$ **do**

4      Play each arm $i \in \hat{\mathcal{D}}(G)$ for $\frac{\alpha' \log(\frac{K}{\Delta_{s+1}})}{\Delta_s^2}$ rounds

5      Update $t$ and $s$.

6 **while** $t \leq T$ **do**

7      Compute a (approximate) solution $x^*_{LP3}$ to LP3.

8      **for** $i \in [K]$ **do**

9          **if** $x^*_{LP3,i} < 1$ **then**

10              **if** $B_i = 0$ *or* $\lfloor B_i + x^*_{LP3,i} \rfloor \geq \lfloor B_i \rfloor$ **then**

11                  Play $i$ and update $t$

12              $B_i += x^*_{LP3,i}$

13          **else**

14              Play $i$ for $\lceil (x^*_{LP3})_i \rceil$ rounds and update $t$

15      Update the phase $s += 1$.

---

$x^*_{LP3,i} \geq 1$, the following inequality holds: $\lceil x^*_{LP3,i} \rceil \leq 2x^*_{LP3,i}$, and thus playing such arms will only increase the incurred regret by a multiplicative factor of at most 2. Thus, we only need to consider $x^*_{LP3,i} < 1$. We introduce a buffer $B \in \mathbb{R}^K$ which will inform us when to play an arm $i$ for which $x^*_{LP3,i} < 1$. The first time the solution of the LP informs us to play $i$ for less than a single round, we play $i$ for a single round and update the buffer as $B_i += x^*_{LP3,i}$. We observe that we have now overplayed $i$ and have a buffer of $1 - x^*_{LP3,i}$ extra plays of $i$. Thus, at the next phase at which $x^*_{LP3,i} < 1$, we can check if $x^*_{LP3,i}$ can be covered by the remaining buffer. If so, then there is no need to play arm $i$ again as we still have sufficient number of observations provided by playing $i$. If the buffer is exceeded, we again play $i$ for one round and take into account the additional overplay. Thus, at the end of phase $s$, the total number of arm $i$ has been played does not exceed

$$2 \sum_{t=1}^{s} x^*_{LP3,i}(t) + \lceil B \rceil \leq 2 \sum_{t=1}^{s} x^*_{LP3,i}(t) + s \leq 3 \sum_{t=1}^{s} x^*_{LP3,i}(t),$$

where $x^*_{LP3,i}(t)$ is the solution to LP3 at phase $t$. The above implies the following lemma.

**Lemma B.1.** *Let $x^*_{LP3,i}(t)$ be the solution to LP3 at phase $t$. The at the end of phase $s$ of Algorithm 2 the total number of plays of arm $i$ is at most*

$$3 \sum_{t=1}^{s} x^*_{LP3,i}(t).$$

The above lemma implies that we can, at the price of a constant multiplicative factor of 3, consider the solution of LP3 instead of the rounded solution played by Algorithm 1. Hence, for the rest of the appendix, we do so.

## B.2 Proof of Theorem 5.2

We begin with a somewhat standard concentration result.

**Lemma B.2.** *For any $s \in [10, \log(T)], K \geq 2, \alpha' \leq 3072$, the following inequality holds*

$$\mathbb{P}(\exists i \in [K] : |\mu_i - \hat{r}_i(s)| \geq b_i(s)) \leq \left(\frac{\Delta_{s+1}}{K}\right)^{\alpha-1}.$$

*Proof.* We use Theorem 1 from Zhao et al. [2016] which states that for a sum of zero-mean, $1/2$ sub-Gaussian random variables $(X_i)_{i=1}^t$ the following inequality holds

$$\mathbb{P}\left(\exists t : \sum_{i=1}^t X_i \geq \sqrt{t(2 \log \log_2(t) + \log(1/\delta))}\right) \leq 2\delta.$$

We begin by bounding $\mathbb{P}(|\mu_i - \hat{r}_i(s)| \geq b_i(s))$ for a fixed $i \in [K]$. Since action $i$ is observed at most $\frac{\alpha' \log(K/\Delta_{s+1})}{\Delta_{s+1}^2}$ times up to and including phase $s$, we can write

$$\mathbb{P}(|\mu_i - \hat{r}_i(s)| \geq b_i(s)) \leq 2\mathbb{P}\left(\exists t \in \left[\frac{\alpha' \log(K/\Delta_{s+1})}{\Delta_{s+1}^2}\right] : \sum_{\ell=1}^t (r_{t,i} - \mu_i) \geq \sqrt{3\alpha t \log\left(\frac{K}{\Delta_{s+1}}\right)}\right)$$

$$\leq 2\left(\frac{\Delta_{s+1}}{K}\right)^\alpha,$$

where we used the fact that for $s \geq 7, K \geq 2, \alpha' \leq 512$ the following inequality holds

$$\log_2\left(\frac{\alpha' \log(K/\Delta_{s+1})}{\Delta_{s+1}^2}\right) \leq \frac{K}{\Delta_{s+1}}.$$

A union bound over $i \in [K]$ completes the proof. $\qquad\square$

**Lemma B.3.** *Under the same assumptions as in Lemma B.2, we have*

$$\mathbb{P}(\exists i \in [K], t \in [s/2, s] : |\mu_i - \hat{r}_i(t)| \geq b_i(t)) \leq \left(\frac{\Delta_{s/2+1}}{K}\right)^{\alpha-2}.$$

*Furthermore, if we let $\mathcal{E}_{upper} = \{\forall i \in [K], t \in [s/2, s] : \hat{\Delta}_i(t) \leq \Delta_i \vee \Delta_t\}$ then $\mathbb{P}(\mathcal{E}_{upper}) \geq 1 - \left(\frac{\Delta_{s/2+1}}{K}\right)^{\alpha-2}$.*

*Proof.* A union bound over Lemma B.2, together with picking $\alpha$ sufficiently large imply

$$\mathbb{P}(\exists i \in [K], t \in [s/2, s] : |\mu_i - \hat{r}_i(t)| \geq b_i(t)) \leq \sum_{t=s/2}^s \left(\frac{1}{2^{t+1} K}\right)^{\alpha-1} \leq \left(\frac{\Delta_{s/2+1}}{K}\right)^{\alpha-2}.$$

For the second part of the lemma, we assume WLOG $\Delta_i \geq \Delta_{s/2}$. Thus, we have

$$\hat{\Delta}_i(t) = \max_{j \in [k]} \hat{r}_j(t) - b_j(t) - \hat{r}_i(t) - b_i(t) =: \hat{r}_{i_t^*}(t) - b_{i_t^*}(t) - \hat{r}_i(t) - b_i(t).$$

On the event that $\{\forall i \in [K], t \in [s/2, s] : |\mu_i - \hat{r}_i(t)| \leq b_i(t)\}$ we have $\hat{r}_{i_t^*}(t) - b_{i_t^*}(t) \leq \mu_{i_t^*}$ and $\hat{r}_i(t) + b_i(t) \geq \mu_i$. This implies that w.p. $1 - \left(\frac{\Delta_{s/2+1}}{K}\right)^{\alpha-2}$ we have

$$\hat{\Delta}_i(t) \leq \mu_{i_t^*} - \mu_i \leq \mu_{i^*} - \mu_i = \Delta_i,$$

for all $i \in [K]$ and $t \in [s/2, s]$. $\qquad\square$

**Lemma B.4.** *Let $\alpha' = 256\alpha$ in LP3. On the event $\mathcal{E}_{upper}$ it holds that $b_i(s) \leq \frac{\Delta_i \vee \Delta_s}{8}, \forall i \in [K]$. Thus, for any $\alpha \geq 3$ and any $\log(K) \vee 10 \leq s \leq \log(T)$ the following inequality holds*

$$\mathbb{P}\left(\exists i \in [K] : b_i(s) \geq \frac{\Delta_i \vee \Delta_s}{8}\right) \leq \left(\frac{\Delta_{s/2+1}}{K}\right)^{\alpha-2}.$$

*Proof.* Recall that $b_i(s) = \sqrt{\frac{3\alpha \log(\frac{K}{\Delta_{s+1}})}{n_i(s)}}$ so we are going to bound $n_i(s)$ from below. Assume that $\Delta_i \geq \Delta_s$, the other case is handled similarly. Let $s_i$ be the phase at which $\Delta_{s_i} = \Delta_i$. On the event $\mathcal{E}_{upper}$ we know that $\hat{\Delta}_i(t) \leq \Delta_i$ for all $t \in [s_i, s], i \in [K]$. The constraints in LP3 imply

$$n_i(s) \geq \sum_{t=s_i+1}^{s} \frac{\alpha'}{\hat{\Delta}_i^2(t)} + \frac{\alpha' \log(\frac{K}{\Delta_{s_i+1}})}{\Delta_{s_i}^2} \geq \frac{\alpha'(s-s_i)}{\Delta_i^2} + \frac{\alpha' \log(\frac{K}{\Delta_{s_i+1}})}{\Delta_{s_i}^2}.$$

The above implies that

$$b_i^2(s) \leq \frac{\alpha \log(\frac{K}{\Delta_{s+1}})}{\frac{\alpha' \log(\frac{K}{\Delta_{s_i+1}})}{\Delta_{s_i}^2} + \frac{\alpha'(s-s_i)}{\Delta_i^2}} \leq \frac{6(s+1)\alpha}{\frac{\alpha'(s_i+1)}{\Delta_i^2} + \frac{\alpha'(s-s_i)}{\Delta_i^2} + \frac{\alpha' \log(K)}{\Delta_i^2}}$$

For $\alpha' = 768\alpha$ the above implies $b_i(s) \leq \frac{\Delta_i}{8}$. $\qquad \square$

**Lemma B.5.** *Let $\mathcal{E}_{gap}(s) = \{\forall i \in [k] : \frac{\Delta_i}{2} \vee \Delta_s \leq \hat{\Delta}_i(s) \leq \Delta_i \vee \Delta_s\}$. Under the assumptions of Lemma B.4 we have*

$$\mathbb{P}\left(\mathcal{E}_{\mathrm{gap}}(s)\right) \geq 1 - 3\left(\frac{\Delta_{s/2+1}}{K}\right)^{\alpha-2}.$$

*Proof.* If $s$ is such that $\Delta_i \leq \Delta_s$ the statement of the lemma holds from Lemma B.3 and the definition of $\hat{\Delta}_i(t)$. We now consider the case $\Delta_i \geq \Delta_s$ and assume that $\mathcal{E}_{upper}$ holds. Lemma B.4 now implies that $b_i(s) \leq \frac{\Delta_i}{8}$. Further, assume that $|\mu_i - \hat{r}_i(s)| \leq b_i(s), \forall i \in [K]$. We have

$$\hat{\Delta}_i(s) = \max_{j \in [K]} \hat{r}_j(s) - b_j(s) - \hat{r}_i(s) - b_i(s) \geq \hat{r}_{i^*}(s) - b_{i^*}(s) - \hat{r}_i(s) - b_i(s)$$

$$\geq \Delta_i - 2(b_{i^*}(s) + b_i(s)) \geq \frac{\Delta_i}{2}.$$

Our assumptions fail with probability at most $\left(\frac{\Delta_{s/2+1}}{K}\right)^{\alpha-2} + 2\left(\frac{\Delta_{s+1}}{K}\right)^{\alpha-1}$. $\qquad \square$

**Lemma B.6.** *For any phase $s \geq \log(K) \vee 10$, it holds that $\mathbb{P}(\hat{\Gamma}_{s+1} \not\subseteq \Gamma_s) \leq 3\left(\frac{\Delta_{s/4+1}}{K}\right)^{\alpha-2}$.*

*Proof.* We have $\mathbb{P}(\hat{\Gamma}_{s+1} \not\subseteq \Gamma_s) \leq \sum_{i \notin \Gamma_s} \mathbb{P}(i \in \hat{\Gamma}_{s+1})$. The fact $i \notin \Gamma_s$ implies that $\Delta_i \geq 2\Delta_s$. The result now follows by Lemma B.5. $\qquad \square$

**Lemma B.7** (Lemma 5.1). *Let $D_{LP3}(s)$ be the value of LP3 at phase $s$ and let $D_{LP4}(s)$ be the value of LP4 at phase $s$. For any $s \geq \log(K) \vee 10$ the following inequality holds*

$$D_{LP3}(s+1) \leq 4D_{LP4}(s),$$

*with probability at least $1 - 3\left(\frac{\Delta_{s/2+1}}{K}\right)^{\alpha-2}$. Further, for any $s \geq \log(K) \vee 10 \vee \log\left(\frac{|I^*|}{\Delta_{\min}}\right)$ we have that the regret incurred for playing according to LP3 is at most $16\alpha' c^*(G, \mu)$ with probability at least $1 - 3\left(\frac{\Delta_{s/2+1}}{K}\right)^{\alpha-2}$.*

*Proof of Lemma 5.1.* For any $s$ and all $i \in [K]$ Lemma B.5 and Lemma B.6 imply that $\hat{\Gamma}_{s+1} \subseteq \Gamma_s$ and $\frac{\Delta^s}{2} \leq \hat{\Delta}(s) \leq \Delta^s$ with probability at least $1 - 3\left(\frac{\Delta_{s/2+1}}{K}\right)^{\alpha-2}$. If we let $x_{LP4}^*(s)$ be a solution to LP4 at phase $s$, then these conditions imply that $4x_{LP4}^*(s)$ is feasible for LP3. This implies

$$D_{LP3}(s+1) \leq 4\langle x_{LP4}^*(s), \hat{\Delta}(s)\rangle \leq 4\langle x_{LP4}^*(s), \Delta^s\rangle = 4D_{LP4}(s).$$

Further, for $s \geq \log(1/\Delta_{\min})$, $\Gamma_s$ consists only of $I^*$. Let $x_{LP3}^*$ be a solution to LP3, and let $\hat{x}^*$ be a solution to the LP dropping all constraints on $I^*$ and its neighborhood. Note that $\langle \hat{x}^*, \hat{\Delta}(s)\rangle \leq 8\alpha' c^*(G, \mu)$ under $\mathcal{E}(s)$. We show by contradiction that

$$\sum_{i \notin I^*} x_{LP3,i}^* \hat{\Delta}_i(s) \leq 2 \sum_{i \notin I^*} \hat{x}_i^* \hat{\Delta}_i(s),$$

which by $\Delta_i \le 2\hat{\Delta}_i(s)$ completes the proof. Assume the opposite is true, take a new $x$ such that

$$x_i = \hat{x}_i^* \ \forall i \notin I^*$$
$$x_i = x_{LP3,i}^* + \sum_{j \notin I^*} x_{LP3,j}^* \ \forall i \in I^* \,.$$

$x$ is a feasible solution of LP3. Next, we only consider $s \ge \log(|I^*|/(4\Delta_{\min}))$ which implies that

$$\langle x, \hat{\Delta}^s(s) \rangle = \sum_{i \in I^*} x_{LP3,i}^* \Delta_s + \sum_{i \notin I^*} (x_{LP3,i}^* |I^*| \Delta_s + \hat{x}_i^* \hat{\Delta}_i(s))$$

$$< \sum_{i \in I^*} x_{LP3,i}^* \Delta_s + \sum_{i \notin I^*} (x_{LP3,i}^* (\frac{\Delta_{\min}}{4} + \frac{\hat{\Delta}_i(s)}{2})$$

$$\le D_{LP3}(s) \,,$$

which is a contradiction to $\langle x, \hat{\Delta}^s(s) \rangle \ge D_{LP3}(s)$.

$\square$

Denote the value of LP2 at phase $s$ as $D_{LP2}(s)$ and a solution to the LP as $x_{LP2}^*(s)$. We note that for any $s$ it holds that $x_{LP4}(s) = (\alpha \log(K/\Delta_{s+1}) \vee \alpha') \sum_{t \le s} x_{LP2}^*(t)$ is feasible for LP4. Further we have that for all $t \le s$ it holds that $\langle x, \Delta^s \rangle \le \langle x, \Delta^t \rangle$. These two observations imply

$$D_{LP4}(s) \le (\alpha \log(K/\Delta_{s+1}) \vee \alpha') \sum_{t \le s} D_{LP2}(t) \,.$$

Further, we have $D_{LP2}(t) \le D_{LP4}(s)$. We can assume that $s \ge 10$, otherwise the regret is $O(K)$. Thus we can characterize the optimality of Algorithm 1 up to factors of $\log^2(1/\Delta_{\min})$ as follows.

**Theorem B.8** (Theorem 5.2). *Let* $d^*(G,\mu) = \max_{s \le \log(|I^*|/\Delta_{\min})} D_{LP2}(s)$. *The expected regret* $R(T)$ *of playing according to Algorithm 2 with* $\alpha = 4$ *and* $\alpha' = 768\alpha$ *is bounded as*

$$R(T) \le O\left(\log^2\left(\frac{1}{\Delta_{\min}}\right) d^*(G,\mu) + \log(T)c^*(G,\mu) + \gamma(G)K\right) \,.$$

*Further, for any algorithm, there exists an environment on which the expected regret of the algorithm is at least* $\Omega(d^*(G,\mu))$.

*Proof.* Lemma 5.1 implies that the regret bounds fail to hold at any phase $s \ge \log(K)$ w.p. at most $3\left(\frac{1}{2^{s/2+1}K}\right)^{\alpha-2}$. Further the regret at phase $s$ is always bounded by $\alpha'K2^s \log(1/\Delta_{min})$ Choosing $\alpha = 4$ implies expected regret of only $O(\log(1/\Delta_{\min}))$ on the union bound of failure events. For the remainder of the proof we now have for $s \le \log(|I^*|/\Delta_{\min})$

$$D_{LP3}(s) \le 4D_{LP4}(s) \le O(\log(K/\Delta_{s+1})d^*(G,\mu)) \,.$$

For $s \ge \log(|I^*|/\Delta_{\min})$ we have that

$$D_{LP3}(s) \le O(c^*(G,\mu)) \,.$$

Finally the regret incurred in the first $s \le \log(K)$ phases is at most $O(\gamma(G)K\log(K))$ as the algorithm plays the approximate solution corresponding to the minimum dominating set of $G$. Combining all of the above shows the instance dependent regret upper bound.

To show the instance independent bound we again use the bound $D_{LP4}(s) \le (\alpha \log(K/\Delta_{s+1}) \vee \alpha') \sum_{t \le s} D_{LP2}(t)$. Let $I_s$ be a maximum independent set of $\Gamma_s$. If $\Delta_s \ge \sqrt{\frac{|I_s|}{T}}$ then by the argument in Section D.1 it holds that $D_{LP2}(s) \le \sqrt{\alpha(G)T}$. Next, we consider $\Delta_s < \sqrt{\frac{|I_s|}{T}}$. Let $s$ be the smallest index for which $\Delta_s < \sqrt{\frac{|I_s|}{T}}$ so that $\Delta_{s-1} \ge \sqrt{\frac{|I_{s-1}|}{T}}$. Since $\Delta_s = \frac{1}{2}\Delta_{s-1}$ we have that $D_{LP2}(s) \le 4D_{LP2}(s-1) \le 4\sqrt{\alpha(G)T}$. Further, playing every arm in $I_s$ for $\frac{1}{\Delta_s^2}$ rounds will exceed the time horizon of the game and incur at most $4\sqrt{\alpha(G)T}$ regret. Suppose that the solution of LP2 at epoch $s$ is such that the game does not end if the solution is followed and consider playing according

LP2 at epoch $s + 1$. If the game does not end, then the incurred regret is no more than $2\sqrt{\alpha(G)T}$. This follows from the fact that

$$\Delta_{s+1} \sum_i x^*_{LP2,i}(s+1) \le \Delta_s \sum_i x^*_{LP2,i}(s+1) \le \Delta_s \sum_{i \in I_s} \frac{4}{\Delta^2_{s-1}} = 2\sqrt{\alpha(G)T},$$

where the last inequality follows from the fact that $x_i = \frac{4}{\Delta^2_{s-1}}, i \in I_s$ is feasible for LP2 at epoch $s$. Proceeding by induction, it holds that any epoch $s$ at which the solution of $LP2$ does not end the game must have regret at most $2\sqrt{\alpha(G)T}$. Suppose now that the game ends, then the regret of the final epoch is at most twice the preceding epochs regret as playing 4-times the optimal solution for the previous epoch is feasible for the current epoch, and so the total regret incurred for playing according to LP2 is at most $4\sqrt{\alpha(G)T}$. By the argument that $D_{LP4}(s) \le (\alpha \log(K/\Delta_{s+1}) \vee \alpha') \sum_{t \le s} x^*_{LP2}(t)$, we can construct a feasible solution for LP4 at any epoch $s$ during which from the solutions to LP2. These feasible solutions have bounded regret for any epoch before the terminal epoch for $LP2$. Finally, we note that the terminal epoch for $LP2$ must be no smaller than the terminal epoch for LP4, otherwise we can construct a feasible solution to LP2 from that of $LP4$ with regret smaller than the value of $LP2$, which would be a contradiction. The construction sets all variables not in $\Gamma_s$ to 0 and rescales the non-zero variables normalizing with the additional logarithmic factors in the constraint of LP4. Thus, the total regret for playing according to $LP4$ is at most $O\left(\log^2(KT)\sqrt{\alpha(G)T}\right)$.

Further, any feasible solution for LP4 is feasible for LP2 and any feasible solution for LP2 must play more than $T$ rounds

The regret lower bound follows from Lemma 6.1. □

## C  Regret lower bounds

### C.1  Proof of Lemma 6.1

**Lemma C.1** (Lemma 6.1). *Fix any instance $\mu$ s.t. $\mu_i \le 1 - 2\Delta_s, i \in I^*$. Let $\Lambda_s(\mu)$ be the set of problem instances with means $\mu' \in \mu + [0, 2\Delta^s]^k$. Then for any algorithm, there exists an instance in $\Lambda_s(\mu)$ such that the regret is lower bounded by LP2.*

*Proof.* We take as a base environment the instance with expected rewards vector $\mu$ and assume that the rewards follow a Gaussian with variance $\frac{1}{\sqrt{2}}$. Let $\tau \in \mathbb{R}_+ \bigcup \{\infty\}$ be the time at which the following is satisfied

$$\min_{i \in \Gamma_s} \mathbb{E}\left[\sum_{t=1}^\tau \mathbb{P}[A_t \in N_i]\right] = \frac{1}{4\sqrt{2}(\Delta_s)^2},$$

where the expectation is with respect to the randomness of the sampling of the rewards and $\mathcal{A}$.

First we argue that we can assume $\tau < \infty$. Consider $\tau = \infty$. Let

$$i^* = \operatorname*{argmin}_{i \in \Gamma_s} \mathbb{E}\left[\sum_{t=1}^\tau \mathbb{P}_\mu[A_t \in N_i]\right].$$

Fix a time horizon $T$ and let $x_T$ be the vector of expected number of observations of $\mathcal{A}$ on environment $\mu$ and $X_T$ the random vector of actual observations. By the assumption that $\tau = \infty$ and Markov's inequality, we have that $\mathbb{P}[X_{T,i^*} \ge \frac{1}{\Delta^2_s}] \le \frac{1}{2}$. Consider the algorithm $\bar{\mathcal{A}}$ which after $\frac{1}{\Delta^2_s}$ observations of $i^*$ switches to playing uniformly at random from $[K] \setminus N_{i^*}$ so that it never observes $i^*$ again. Let $\mu'$ be the instance which changes the expected reward of $\mu_{i^*}$ to $\mu'_{i^*} = \mu_{i^*} + 2\Delta_s$, and so $\mu' \in \Lambda_s(\mu)$. The KL-divergence between the measures induced by playing $\mathcal{A}$ on these two instances for $\tau$ rounds is bounded as $4\Delta^2_s x_{\tau,i^*}$. If we let $x'_T$ denote the vector of expected number of observations under environment $\mu'$ then Pinsker's inequality implies that

$$x'_{T,i^*} \le \frac{1}{4\sqrt{2}\Delta^2_s} 2\Delta_s\sqrt{x_{\tau,i}/2} \le \frac{1}{4\Delta^2_s}.$$

Thus, the expected regret of $\bar{\mathcal{A}}$ under $\mu'$ is at least $(T - \frac{1}{4\Delta^2_s})\Delta_s$ for any $T \ge \frac{1}{4\Delta^2_s}$. Further, by Pinsker's inequality the probability that $X_{T,i^*} \ge \frac{1}{\Delta^2_s}$ under $\bar{\mathcal{A}}$ in environment $\mu'$ is bounded by $\frac{3}{4}$.

Since $\mathcal{A}$ and $\bar{\mathcal{A}}$ act in the same way up to $\frac{1}{\Delta_s^2}$ observations of $i^*$ it holds that the expected regret of $\mathcal{A}$ in environment $\mu'$ is at least $\frac{1}{4}(T - \frac{1}{4\Delta_s^2})\Delta_s$ for any $T \geq \frac{1}{4\Delta_s^2}$. Thus for $T$ large enough, e.g., $T = \Omega(D_{LP2(s)}/\Delta_s + \frac{1}{4\Delta_s^2})$ the conclusion of the lemma holds.

We now assume that $\tau \leq \infty$. Let $x_{\tau,i} = \mathbb{E}\left[\sum_{t=1}^\tau \mathbb{P}[A_t \in N_i]\right]$ be the expected number of observations of action $i$ after $\tau$ rounds. Assume that $\sum_{i \notin \Gamma_s} x_{\tau,i}\Delta_i \leq \frac{D_{LP2}(s)}{16}$, otherwise we are done. The definition of $\tau$ with the above assumption imply that

$$x_{\tau,i} \geq \frac{1}{4\sqrt{2}\Delta_s^2}, \forall i \in \Gamma_s$$

$$\Longrightarrow$$

$$\sum_{i\in[K]} x_{\tau,i}\Delta_i^s \geq \frac{\bar{D}(\Delta^s, \Gamma_s)}{4\sqrt{2}}$$

$$\Longrightarrow$$

$$\tau\Delta_s \geq \sum_{i\in\Gamma_s} x_{\tau,i}\Delta_s \geq \frac{\bar{D}(\Delta^s, \Gamma_s)}{16}.$$

Let $\mu'$ be the instance which changes the expected reward of $\mu_{i^*}$ to $\mu'_{i^*} = \mu_{i^*} + 2\Delta_s$, and so $\mu' \in \Lambda_s(\mu)$. The KL-divergence between the measures induced by playing $\mathcal{A}$ on these two instances for $\tau$ rounds is bounded as $4\Delta_s^2 x_{\tau,i^*}$. If we let $x'_\tau$ denote the vector of expected number of observations under environment $\mu'$ then Pinsker's inequality implies that

$$x'_{\tau,i^*} \leq \tau\Delta_s\sqrt{x_{\tau,i}/2} \leq \frac{\tau}{2}.$$

This implies

$$\sum_{i\in[K]\setminus\{i^*\}} x'_{\tau,i}\Delta_i^s \geq \frac{\tau}{2}\Delta_s \geq \frac{\bar{D}(\Delta^s, \Gamma_s)}{32}.$$

$\square$

## C.2    Proof of Theorem 4.1

**Theorem C.2.** *There exists a feedback graph $G$, with $K \geq 32$ vertices, such that for any algorithm $\mathcal{A}$ there exists an environment $\mu$ on which $R(T) \geq \Omega(K^{1/8}c^*(G,\mu))$.*

*Proof.* For any algorithm $\mathcal{A}$, define the algorithm $\overline{\mathcal{A}}$ as follows: If there have been more than $\frac{K^{\frac{7}{8}}}{64\Delta_2^2}$ pulls of actions in $\mathcal{N}_2$, then commit to action $\mathcal{N}_3$ until end of time. We call the random time-step where $\mathcal{A}$ and $\overline{\mathcal{A}}$ deviate in trajectory as $\tau$. Define the stopping times

$$T_1 := \min\left\{t \in \mathbb{N} \cup \{\infty\} \,\middle|\, \mathbb{P}[\tau \leq t] > \frac{1}{2}\right\}$$

$$T = \min\left\{\left\lceil\frac{K}{128\Delta_2^2}\right\rceil, T_1\right\}.$$

Let $n^*$ be the node in $\mathcal{N}_2$ with the smallest number of expected observations at time $T$ under algorithm $\overline{\mathcal{A}}$. Let $N_i$ denote the number of times an action in $\mathcal{N}_i$ has been played by $\overline{\mathcal{A}}$. The total number of observations over all actions in $\mathcal{N}_2$ is

$$2N_1 + K^{\frac{1}{8}}N_2 \leq \frac{K}{64\Delta_2^2} + \frac{K}{64\Delta_2^2} = \frac{K}{32\Delta_2^2}.$$

Hence the number of observations of $n^*$ is bounded by $\frac{1}{32\Delta_2^2}$. Consider the environment $\mathcal{E}_1$, where all we change is increasing the reward of $n^*$ by up to $2\Delta_2$. By Pinsker's inequality we have

$$|\mathbb{P}_{\overline{\mathcal{A}},\mathcal{E}}(E) - \mathbb{P}_{\overline{\mathcal{A}},\mathcal{E}_1}(E)| \leq \sqrt{\frac{1}{2}(2\Delta_2)^2\frac{1}{32\Delta_2^2}} = \frac{1}{4},$$

as the largest difference in probability of any event under the two environments. We consider two possible cases below.

**Case 1** $T < T_1$**.** Set the reward of $n^*$ to $\nu + \Delta_2$. Define the following event:

$$E := \left\{ \tau > T \wedge N_{n^*} \le \frac{1}{4\Delta_2^2} \right\}.$$

In the first environment, we have

$$\mathbb{P}(E) = 1 - \mathbb{P}(E^C) \ge 1 - \mathbb{P}[\tau < T] - \mathbb{P}\left[ N_{n^*} > \frac{1}{4\Delta_2^2} \right] \ge \frac{3}{8}.$$

Hence the probability of $E$ is at least $\frac{1}{8}$ in the changed environment. The regret of $\mathcal{A}$ is at least

$$\frac{1}{8}\left( T - \frac{1}{4\Delta_2^2} \right)\Delta_2 > \frac{K}{1024\Delta_2} = \Omega\left( \frac{K^{\frac{3}{4}}}{\Delta} \right).$$

However, the value of LP1 for this environment is $\Theta\left( \frac{K^{\frac{7}{8}}}{\Delta_2} \right) = \Theta\left( \frac{K^{\frac{5}{8}}}{\Delta} \right)$.

**Case 2** $T = T_1$    Set the reward of $n^*$ to $\nu$. Define the following event:

$$E := \left\{ \tau \le T \wedge N_{n^*} \le \frac{1}{4\Delta_2^2} \right\}.$$

In the base environment, we have

$$\mathbb{P}(E) = 1 - \mathbb{P}(E^C) \ge 1 - \mathbb{P}[\tau > T] - \mathbb{P}[N_{n^*} > \frac{1}{4\Delta_2^2}\}] \ge \frac{3}{8}.$$

Hence the probability of $E$ is at least $\frac{1}{8}$ in the changed environment. The regret of $\mathcal{A}$ is at least

$$\frac{1}{8}\left( \frac{K^{\frac{7}{8}}}{64\Delta_2^2} - \frac{1}{4\Delta_2^2} \right)\Delta_2 > \frac{K^{\frac{7}{8}}}{1024\Delta_2} = \Omega\left( \frac{K^{\frac{5}{8}}}{\Delta} \right).$$

However, the value of LP1 for this environment is $\Theta\left( \frac{K^{\frac{1}{2}}}{\Delta} \right)$.

We note that this also characterizes $d^* = \Theta(K^{1/8}c^*)$, as playing according to LP2 will result in $\Theta(K^{7/8}/\Delta_2^2)$ plays of arms in $\mathcal{N}_2$.

Hence for any algorithm, there exist an environment and time step $T = \mathcal{O}(\frac{K^{\frac{1}{2}}}{\Delta^2})$, such that the algorithm suffers a regret that is a factor $K^{\frac{1}{8}}$ larger than $c^*$. $\qquad\square$

# D    Characterizing $d^*$

## D.1    Improving on bound in Lykouris et al. [2020]

We now show that $d^*(G, \mu) \le \max_{I \in \mathcal{I}(G)} \sum_{i \in I} \frac{1}{\Delta_i}$:

$$D_{LP2}(s) \le \frac{\gamma(\Gamma_s)}{\Delta_s} \le \frac{\alpha(\Gamma_s)}{\Delta_s} \le \sum_{i \in \mathcal{I}(\Gamma_s)} \frac{1}{\Delta_i} \le \max_{I \in \mathcal{I}(\Gamma_s)} \sum_{i \in I} \frac{1}{\Delta_i} \le \max_{I \in \mathcal{I}(G)} \sum_{i \in I} \frac{1}{\Delta_i}$$

$$\implies d^*(G, \mu) \le \max_{I \in \mathcal{I}(G)} \sum_{i \in I} \frac{1}{\Delta_i}.$$

The first inequality follows from the definition of the LP, the second inequality follows from the fact that the domination number is no larger than the independence number, the third inequality follows from the fact that for any $i \in \Gamma_s$ we have $\Delta_s \ge \Delta_i$, and the fifth inequality holds by the fact that $\mathcal{I}(\Gamma_s) \subseteq \mathcal{I}(G)$.

## D.2 Bound on $d^*$ for star-graphs

**Lemma D.1.** *For the star-graph $G$ and any instance $\mu$, the following inequality holds: $c^* + \frac{|I^*|}{\Delta_{\min}} \geq d^*$.*

*Proof.* Consider the dual of LP1 given below

$$\max_{y \in \mathbb{R}^K} \frac{1}{\Delta_s^2} \sum_{i \in \Gamma_s} y_i$$
$$s.t. \sum_{j \in N_i \cap \Gamma_s} y_j \leq \Delta_s, \forall i \in \Gamma_s, \tag{LP5}$$
$$\sum_{j \in N_i \cap \Gamma_s} y_j \leq \Delta_i, \forall i \in \Gamma_s^{\mathsf{C}}.$$

Note that for any $i \in [K]$ we can take the intersection of $N_i$ with $\Gamma_s$ as no action $j \in \Gamma^{\mathsf{C}}$ can increase the value of the objective of LP5. The analysis is split into two parts. First consider all phases $s$ for which it holds that $\Delta_r \leq \Delta_s$. We argue that the solution to LP2 for these phases is to just play the revealing vertex for $\frac{1}{\Delta_s^2}$ times. Indeed we can just set $y_r = \Delta_s$ and observe that this is feasible for the dual LP with value $\frac{1}{\Delta_s}$. Further, setting $x_r = \Delta_s$ in the primal also yields a value of $\frac{1}{\Delta_s}$. The fact that $\Delta_r \geq \Delta_{\min}$ together with the lower bound of $R(T) \geq \Omega(\frac{1}{\Delta_{\min}})$ for any strategy, implies that playing according to LP2 is optimal up to at least the phase at which $\Delta_r > \Delta_s$.

Next, consider the setting of $s$ s.t. $\Delta_r > \Delta_s$. The following is feasible for LP5

$$y_i = \begin{cases} \frac{\Delta_r}{|\Gamma_s|} & \text{if } |\Gamma_s|\Delta_s \geq \Delta_r \\ \Delta_s & \text{otherwise.} \end{cases}$$

Thus the value of LP2 is $\frac{\Delta_r}{\Delta_s^2}$ in the first case and $\frac{|\Gamma_s|}{\Delta_s}$ in the second as we can match these values in the primal by setting either $x_r = \frac{1}{\Delta_s^2}$ or $x_i = \frac{1}{\Delta_s^2}, i \in \Gamma_s$ in the primal. To show that both of these values are dominated by $c^*$ consider the dual of LP1 below

$$\max_{y \in \mathbb{R}^K} \sum_{i \in [K] \setminus I^*} \frac{y_i}{\Delta_i^2}$$
$$s.t. \sum_{j \in N_i \cap \Gamma_s} y_j \leq \Delta_i \forall i \in [K]. \tag{LP6}$$

First consider the setting in which the value of LP2 equals $\frac{\Delta_r}{\Delta_s^2}$. Set all $y_i \in \Gamma_s \setminus I^*$ to $y_i = \frac{\Delta_r}{|\Gamma_s|}$ and all other $y_i = 0$. This is feasible for LP6 and implies that

$$c^* \geq \sum_{i \in \Gamma_s \setminus I^*} \frac{\Delta_r}{\Delta_i^2 |\Gamma_s|} \geq \frac{|\Gamma_s \setminus I^*|}{|\Gamma_s|} \frac{\Delta_r}{\Delta_s^2},$$

where the second inequality follows because $\Delta_s \geq \Delta_i$. This is sufficient to guarantee that $c^* + \frac{|I^*|}{\Delta_{\min}} \geq \frac{\Delta_r}{\Delta_s^2}$. Next consider the setting in which the value of LP2 equals $\frac{|\Gamma_s|}{\Delta_s}$. Set all $y_i : i \in \Gamma_s \setminus I^*$ to $y_i = \Delta_i$ and all other $y_i = 0$. This is again feasible for LP6 because $\sum_{i \in \Gamma_s \setminus I^*} \Delta_i \leq |\Gamma_s|\Delta_s \leq \Delta_r$ and further implies that

$$c^* \geq \sum_{i \in \Gamma_s \setminus I^*} \frac{1}{\Delta_i} \geq \frac{|\Gamma_s \setminus I^*|}{|\Gamma_s|} \frac{1}{\Delta_s}.$$

Again this is sufficient to guarantee that $c^* + \frac{|I^*|}{\Delta_{\min}} \geq \frac{|\Gamma_s|}{\Delta_s}$. $\qquad\square$

## D.3 Proof of Lemma 7.2

*Proof.* Again we assume that $G$ consists only of a single connected component. First we argue that $\mathcal{C}(G)$ is a star-graph. Consider three vertices $v_1, v_2, v_3 \in \mathcal{C}(G)$ such that $v_1, v_3 \in N_{v_2}$. Assume that $v_1 \in N_{v_3}$. This implies that there exists a vertex $u \in \mathcal{C}(G)$ such that $u \in N_i$ but $u \notin N_j$ for $i \neq j, i, j \in \{1, 2, 3\}$, otherwise $N_{v_1} = N_{v_2} = N_{v_3}$ and they collapse to a single vertex under $\mathcal{C}(G)$.

Assume that $u \in N_1$ but $u \notin N_2$. Then this implies there exists a path of length 3 between $u$ and $v_2$, given by $(u, v_1, v_3, v_2)$. All other cases are symmetric and so this contradicts $v_2 \in N_{v_3}$. Further, it can not occur that there exists a neighbor $u$ of $v_2$ or $v_3$ s.t. $u \notin N_{v_1}$. The above two arguments show that for $G$ every vertex must neighbor $v_1$ and no two vertices $v_2, v_3 \neq v_1$ can be neighbors making $\mathcal{C}(G)$ a star graph.

Next, we show that for any $\mu$ there exists a $\mu'$ defined on $\mathcal{C}(G)$ s.t. $c^*(G, \mu) = c^*(\mathcal{C}(G), \mu')$ and $d^*(G, \mu) = d^*(\mathcal{C}(G), \mu')$. For any equivalence class $[v] \in \mathcal{C}(G)$, define the expected reward of $[v]$ as $\mu'_{[v]} = \max_{u \in [v]} \mu_u$. For the remainder of the proof we represent the equivalence class by the action $v$ with maximum reward $\mu_v = \mu'_{[v]}$. By construction, $\max_v \mu'_{[v]} = \max_u \mu_u$, hence the gaps are also identical $\Delta_{[v]} = \Delta_v$. We first show that we can drop the constraints for any $u \in [v] \setminus \{v\}$ without changing the value of the LP. The LHS of all constraints for $u \in [v]$ is identical since it depends only on $N_u$. Hence, we can remove all but the largest constraint, which is obtained for the smallest gap, i.e. the constraint for $v$. Next we show that we can also remove $x_u$ for any $u \in [v] \setminus \{v\}$. Assume $x_u > 0$ is a feasible solution of the LP, then we obtain another feasible solution $x'$ by $x'_u = 0, x'_v = x_u + x_v$, while leaving everything else unchanged. However, since $\Delta_v \leq \Delta_u$, the objective value of $x'$ is smaller or equal that of $x$. Hence there exists an optimal solution where all $u \in [v] \setminus \{v\}$ are $0$ and these variables can be dropped from the LP. The resulting LP after dropping constraints and variables for $u \in [v] \setminus \{v\}$ is exactly given by $\mathcal{C}(G), \mu'$. This shows that $c^*(\mathcal{C}(G), \mu') = c^*(G, \mu')$.

The claim that $d^*(G, \mu) = d^*(\mathcal{C}(G), \mu')$ follows analogously.

$\square$