# OpenReview forum: "Stochastic Online Learning with Feedback Graphs: Finite-Time and Asymptotic Optimality"
_NeurIPS.cc/2022/Conference — NeurIPS 2022 Accept_

### Official Review · Reviewer_zGSQ · 2022-07-11

**Rating:** 6
**Confidence:** 1
**Soundness:** 3 good
**Presentation:** 3 good
**Contribution:** 3 good

**Summary:**

This paper investigates the optimality in the problem of stochastic online learning with feedback graphs.
The authors showed that in the problem setting, the vanilla asymptotic optimality and finite-time guarantee cannot be achieved simultaneously.
The authors provide a novel notion of finite-time complexity and propose an algorithm with which the quasi-optimal regret bound can be obtained both asymptotically and in finite time.

**Questions:**

To obtain similar results, can we remove the assumptions of undirectedness of the feedback graph and self-loops at every vertex?

**Strengths And Weaknesses:**

Strengths:
- The paper discovered the surprising fact that the vanilla asymptotic optimality and finite-time guarantee cannot be achieved simultaneously, and a good amount of explanation is provided to explain the intuition behind it.
- The paper is well written, and easy to follow the arguments.


Weaknesses:
- The assumption introduced in this paper might be a bit strong, while a sufficient amount of interesting contribution is provided. For example, if we assume self-loops on the feedback graphs, stochastic online learning with feedback graphs is intrinsically easier than the multi-armed bandits. A more detailed comparison of assumptions against existing work might be desired.

(minor)
There are typos:
line 146, 148: $\Delta_{min}$ should be $\Delta_{\min}$ and same for $\Delta_{max}$.
line 162: 1/e should be multiplied?

---

> ### Author Response · Authors · 2022-08-02
> **Response to reviewer zGSQ**
>
> > The assumption introduced in this paper might be a bit strong, while a sufficient amount of interesting contribution is provided. For example, if we assume self-loops on the feedback graphs, stochastic online learning with feedback graphs is intrinsically easier than the multi-armed bandits. A more detailed comparison of assumptions against existing work might be desired.
>
> > To obtain similar results, can we remove the assumptions of undirectedness of the feedback graph and self-loops at every vertex?
>
> We included the undirected edges assumption to avoid making the presentation even more complex and technical than it already is. That being said, we expect the extension to the directed case to be merely a technical exercise with similar results. Removing self-loops at every vertex is known to yield a very different regime which might require a different algorithm. It is known that the min-max regret rates become $O(\beta^{1/3}T^{2/3})$ in the general partially observable setting (without self-loops) (Alon et al. (2015)). We note that self-loops and undirected edges are standard assumptions also made in the works of Buccapatnam et al. (2014, 2017), Wu et al. (2015), Li et al. (2020), Lykouris et al. (2020).

---

### Official Review · Reviewer_Ur6a · 2022-07-12

**Rating:** 7
**Confidence:** 2
**Soundness:** 3 good
**Presentation:** 3 good
**Contribution:** 3 good

**Summary:**

The paper considers the problem of stochastic online learning with feedback graphs. This is essentially the classic stochastic multi-armed bandit problem but with a more intricate feedback model that allows for additional side information. Specifically, given a graph whose vertices represent the arms, querying an arm reveals a sample of its reward as well as that of all of its neighbors in the graph. This setting has already seen significant research and the author's contributions are:
1. Showing that several existing algorithms do not achieve the asymptotic lower bound or have a significant finite time overhead
2. Showing a graph G for which any algorithm performs much worse than the asymptotic rate
3. Showing a new quantity $d^*$ that lower bounds the performance of any algorithm
4. An algorithm with quasi-optimal regret with respect to the quantity $d^*$ and the asymptotic rate.

**Questions:**

See in the strengths and weaknesses section.

**Limitations:**

Yes

**Strengths And Weaknesses:**

The paper considers the issue of finite time guarantees of MAB under graph feedback. The main claims are interesting and point out shortcomings in the existing literature. There are also novel ideas both in the algorithm and the lower bounds. Overall the paper is well-written and has solid contributions.

I do have some comments/concerns:
1. On the face of it, Theorem 4.1 seems to contradict previous upper bounds. It's supposed to show that finite time performance may be much worse than that of the asymptotic regime. However, there doesn't seem to be a problem applying the result asymptotically to conclude that $c^*$ is not the asymptotic bound but at least $K^{1/8} c^*$. The results cited by the authors achieve the asymptotic rate (maybe under some assumptions?) and so some thing doesn't add up. Can you explain this? This should probably be explained in the paper.
2. Also regarding Theorem 4.1, Why is the conclusion that the finite time regime is $T \le O(\exp(K^{1/8}))$? The $K^{1/8}$ multiplies both terms in the regret.
3. While there is an extensive comparison to previous works, it is somewhat hard to grasp the larger picture. It seems that previous methods and lower bounds have some advantages and some disadvantages but I found it hard to fully compare. Can you summarize all of these in a table or something similar?
4. The paper ends by showing some cases where $d^*$ is relatively small and thus the asymptotic rate is relevant even in finite time. However, I did not find any concrete examples where the opposite occurs, i.e., $d^*$ is dominant up to a relatively large horizon. Perhaps Theorem 4.1 is meant to accomplish this? If so, it is not written in terms of $d^*$. Can you give more concrete examples?


**Typos:**
* line 23: different => difference
* line 116: "work the" delete "the"
* line 121: work => works'
* line 201: delete "the" and "indeed"
* line 210: preferences => preference
* line 358: equivalent => equivalence

---

> ### Author Response · Authors · 2022-08-02
> **Response to reviewer Ur6a**
>
> > On the face of it, Theorem 4.1 seems to contradict previous upper bounds. It's supposed to show that finite time performance may be much worse than that of the asymptotic regime. However, there doesn't seem to be a problem applying the result asymptotically to conclude that $c^*$ is not the asymptotic bound but at least $K^{1/8}c^*$. The results cited by the authors achieve the asymptotic rate (maybe under some assumptions?) and so some thing doesn't add up. Can you explain this? This should probably be explained in the paper.
>
> Please see the corrected statement of Theorem 4.1 in our general comment.
>
> > Also regarding Theorem 4.1, Why is the conclusion that the finite time regime is $T\leq \mathcal{O}(\exp(K^\frac{1}{8}))$? The $K^{1/8}$ multiplies both terms in the regret.
>
> Please see the corrected statement of Theorem 4.1 in our general comment.
>
> > While there is an extensive comparison to previous works, it is somewhat hard to grasp the larger picture. It seems that previous methods and lower bounds have some advantages and some disadvantages but I found it hard to fully compare. Can you summarize all of these in a table or something similar?
>
>  We plan to add a table which compares all of the stated bounds on the example presented in Figure 1 (b). To summarize the results $c^* = O(\frac{\log(T)}{\Delta_{\min}})$, the minmax rate is $O(\sqrt{KT})$, the works of Caron et al. (2012), Cohen et al. (2016), Buccpatnam et al. (2014, 2017) all have a regret bound of the order $O(\frac{K\log(T)}{\Delta_{\min}})$. The works of Wu et al. (2015) and Li et al. (2020) have a regret bound of the order $O(\frac{\log(T)}{\Delta_{\min}} + \frac{K}{\Delta_{\min}^2})$ and finally our Theorem 5.2 shows a regret bound of $O(\frac{\log(T)}{\Delta_{\min}})$.
>     \item Theorem 4.1 indeed describes an instance where $d^*$ is lower bounded by $K^{1/8}c^*$. We will add an explicit computation of $d^*$ for the specific graph used in the proof of Theorem 4.1 to the paper.
>
> > The paper ends by showing some cases where $d^*$ is relatively small and thus the asymptotic rate is relevant even in finite time. However, I did not find any concrete examples where the opposite occurs, i.e., $d^*$ is dominant up to a relatively large horizon. Perhaps Theorem 4.1 is meant to accomplish this? If so, it is not written in terms of $d^*$. Can you give more concrete examples?
>
> Theorem 4.1 indeed describes an instance where $d^*$ is lower bounded by $K^{1/8}c^*$. We will add an explicit computation of $d^*$ for the specific graph used in the proof of Theorem 4.1 to the paper.

---

> > ### Comment · Reviewer_Ur6a · 2022-08-09
> > **post rebuttal**
> >
> > I thank the authors for addressing my concerns. With the typo correction, Theorem 4.1 is now sensible. Given the additional clarifications that the authors promise in the final version, I currently plan on keeping my score.

---

### Official Review · Reviewer_WHxr · 2022-07-13

**Rating:** 7
**Confidence:** 3
**Soundness:** 3 good
**Presentation:** 3 good
**Contribution:** 3 good

**Summary:**

This paper considers a general bandit-type learning problem. Arms are represented by nodes. When an arm is played, the feedback is shown for every arm corresponding the vertices adjacent to the vertex that corresponds to the pulled arm. Thus this is more general than full-feedback models as well as bandit models (corresponding to the complete and empty graphs respectively). The authors consider asymptotic regret bounds of the form $c\log T+d$ and show that for any particular \textit{finite time}, the bounds are not very meaningful in some cases. The authors propose a new quantity $d^*$ to design algorithms based upon. This quantity is related to the regret incurred by an algorithm when run on an instance when the means are all within $\Delta$ of each other. A new algorithm is provided whose regret is roughly of the form $c^*\log T + d^*$, and this $d^*$ is shown to be meaningful via a lower bound. To underscore the point, various other approaches are critiqued.

**Questions:**

I think the sufficient condition for Lemma 7.2 is not very clear, perhaps some examples of graphs would be helpful.

Can the second set of constraints in LP3 and LP4 be discussed? They are not present in LP2, and I guess they are useful to control the estimates of arms that have been eliminated?

In page 5: "In this paper, we opt for the second choice...", is a little confusing, since one would think Algorithm B is the second one.

Can it be confirmed that this is the idea: "Different algorithms lead to different parameters $c^*$ and $d$ depending on the instance (for instance, as shown in Table 1). asymptotically, only $c^*$ matters, but for finite time, $d$ matters as well. This paper presents a $d^*$ that is achievable by Alg 1, which only depends on the topology of the graph, and which is necessary for graphs of that topology." In this sense, this is an instance dependent min-max bound, where the "instance" is not the complete instance, but just the topology.

**Limitations:**

I am not sure I understand the paragraph starting at line 206. it establishes that this is just one notion of optimality, perhaps it might be reasonable to try and minimize the competitive ratio between $c^*$ and $d$. I do understand that this is not the direction taken by the paper, and I think the contribution is nice anyway.

**Strengths And Weaknesses:**

Strength: the proof sketches are very helpful. The problem considered is nice; I think in the bandit literature it isn't really discussed and not as important. The comparisons with the other LPs are insightful.

---

> ### Author Response · Authors · 2022-08-02
> **Response to reviewer WHxr**
>
> > I think the sufficient condition for Lemma 7.2 is not very clear, perhaps some examples of graphs would be helpful.
>
> We thank the reviewer for the suggestion and add some example graphs to the paper. In general the graphs will be star-shaped, however, we can replace each vertex by a clique.
>
> > Can the second set of constraints in LP3 and LP4 be discussed? They are not present in LP2, and I guess they are useful to control the estimates of arms that have been eliminated?
>
> We `soft-eliminate' arms after about $\frac{\log(1/\Delta)}{\Delta^2}$ observations. To ensure that we did not falsely eliminate an optimal arm, we need to ensure that eliminated arms still receive at least $\log(t)/(\Delta^2)$ observations over time. The second constraint ensures exactly this.
> An alternative design choice in the algorithm would have been to ``hard-eliminate'' arms dependent on a confidence interval scaling with $\log(T)$. However, we did not pursue this path because it led to additional $\log(T)$ factors in the regret bound.
>
> > In page 5: "In this paper, we opt for the second choice...", is a little confusing, since one would think Algorithm B is the second one.
>
> Thank you, we clarify this point in the revision.
>
> > Can it be confirmed that this is the idea: "Different algorithms lead to different parameters $c^*$ and $d$ depending on the instance (for instance, as shown in Table 1). asymptotically, only $c^*$ matters, but for finite time, $d$ matters as well. This paper presents a $d^*$ that is achievable by Alg 1, which only depends on the topology of the graph, and which is necessary for graphs of that topology." In this sense, this is an instance dependent min-max bound, where the "instance" is not the complete instance, but just the topology.
>
> This statement is partly correct. The quantity $c^*$ depends only on the topology and the gaps, but not on the algorithm. $d$ is indeed an algorithm-dependent quantity. We present an algorithm where $d$ is upper bounded by $d^*$, which depends on both the topology and the gaps.
> In our lower bound regarding $d^*$, we show that for every instance (defined by  (topology,mean rewards)), there exists an instance with the same topology and ``similar'' mean rewards such that the regret is lower bounded by $\tilde{\Omega}(d^*)$.
> It is not exactly a minimax bound because $d^*(G,\mu')$ for $\mu'$ being a "similar" environment, can differ from the value $d^*(G,\mu)$ for the original problem instance.
>
> > I am not sure I understand the paragraph starting at line 206. it establishes that this is just one notion of optimality, perhaps it might be reasonable to try and minimize the competitive ratio between $c^*$ and $d$. I do understand that this is not the direction taken by the paper, and I think the contribution is nice anyway.
>
> The paragraph starting at line 206 indeed seeks to motivate our choice of the notion of optimality. We argue that minimizing the competitive ratio between $c^*$ and $d$ is not as well aligned with the idea of regret minimization and that our choice can be more compatible with regret minimization.

---

### Official Review · Reviewer_D2tm · 2022-07-13

**Rating:** 4
**Confidence:** 2
**Soundness:** 2 fair
**Presentation:** 2 fair
**Contribution:** 2 fair

**Summary:**

The paper considers the important  problem of stochastic bandits on a graph: Given a known graph (V,E), there are unknown reward distributions with each vertex. At each stage, the learner can choose a vertex. Random rewards are then sampled. The learner accumulates the reward associated with their chosen arm (vertex), but also observes the reward of all vertices with an adjoining edge.

The primary focus of the present paper is on gap-dependent bounds which hold in finite-time, but are also asymptotically optimal.

After introducing the problem, the paper turns to addressing concerns regarding existing methods and associated regret bounds. In particular, it is shown that the UCB-LP based approach of Buccapatnam et al. can yield regret which is not asymptotically optimal for certain graphs, such as star graphs, where the method doesn’t optimally take advantage of the structure. Also considered are the more refined approaches of Li et al. 2020 which is asymptotically optimal, but has quite severe dependence on the minimal gap.

The central result of the next section is Theorem 4.1, which gives an instance dependent lower bound. There is also some discussion of the complexities surrounding optimal bounds in the bandit case compared with the other two extremes of regular bandit and full information.

Section 5 turns to positive results and presents a new algorithm. The algorithm is, broadly speaking, a careful refinement of the UCB-LP methodology. In addition, a regret bound is presented in Theorem 5.2 for the algorithm. The regret bound is asymptotically optimal and has a less severe dependence upon the minimal gap than Li et al.’s bound.

The regret bound in Theorem 5.2 contains a new quantity “d*”. Section 6 includes a lower bound which shows that this dependence is in some sense necessary. Section 7 continues the comparison with other methods and shows that the quantity d* is always exceeded by key terms in the regret bound of Lykouris 2020, and can be strictly smaller. Section 6 also includes a sketch of the proof of Theorem 4.1.


**Questions:**

Is it possible to say under what regimes the bound in Theorem 5.2 is minimax optimal up to logarithmic factors in the “gap-free” case?

Can you explain the ordering of the quantifiers in Theorem 4.1? It seems as if the regret lower bound is claimed to hold for a single bandit problem, for all sufficiently large times T. This seems to contradict Theorem 5.2 when K\geq C^8 where C here denotes the constant subsumed in big O notation in Theorem 5.2.


**Limitations:**

The primary limitations are discussed above in the “strengths and weaknesses” section.

**Strengths And Weaknesses:**

The problem of bandits with feedback graphs is of great interest to the Machine Learning community. Moreover, the task of achieving bounds which are asymptotically optimal and have provably strong finite time performance seems of key importance.

A key strength of the paper is the new methodology and associated regret bound in Theorem 5.2. The regret bound is (1) asymptotically optimal and (2) seems to outperform that of Li et al. in the low gap regime.

That said, its important to note that the theory in the present paper appears to leave an incomplete picture. It’s not clear if the bounds are both asymptotically and minimax optimal up to polylog factors in the “gap-free” case.

There also seems to be an issue with Theorem 4.1. It seems as if the regret lower bound is claimed to hold for a single bandit problem, for all sufficiently large times T. This seems to contradict Theorem 5.2 when K\geq r^8 where r here denotes the ratio of the constants subsumed in big O notation in Theorem 5.2, divided by the constant subsumed in the Omega notation of Theorem 4.1.

The discussion in Section 4 also seems to be unnecessarily confusing. Ultimately, specifying optimality requires a combination of algorithm independent lower bounds and as well as upper bounds. The former always necessitates specifying an appropriate class of problem instances. This seems true in general, and for bandits on graphs in particular. Perhaps the point being made here is more subtle, but it remained somewhat elusive to me.

Overall the structure of the paper is strange. The discussion of related work appears in several places in a somewhat confusing fashion. The proof sketch of Theorem 4.1 appears in Section 6 rather than 4. Section 5 should be probably be moved towards the front of the paper. The two lower bounds could be placed together in a common section.

---

> ### Author Response · Authors · 2022-08-02
> **Response to reviewer D2tm**
>
> > Is it possible to say under what regimes the bound in Theorem 5.2 is minimax optimal up to logarithmic factors in the “gap-free” case?
>
> The upper bound given by Theorem 5.2 always matches up to poly-$\log$ factors the gap-free minimax rate of $\sqrt{\alpha T}$. We will add this and the proof to the paper. To see this informally, notice that, at every epoch, playing a maximum independent set of vertices in $\Gamma_s$ for $\tilde\Omega(1/\Delta_s^2)$ in LP~2 is a feasible solution with regret at most $\alpha 2^s$. Since there are at most $\frac{\log_2(T/\alpha)}{2}$ such rounds, we have a total regret of at most $\tilde O(\sqrt{\alpha T})$.
>
> > Can you explain the ordering of the quantifiers in Theorem 4.1? It seems as if the regret lower bound is claimed to hold for a single bandit problem, for all sufficiently large times T. This seems to contradict Theorem 5.2 when $K\geq C^8$ where $C$ here denotes the constant subsumed in big $O$ notation in Theorem 5.2.
>
> Please see the corrected statement of Theorem 4.1, which should resolve the contradiction in our general comment.
>
> >  The discussion in Section 4 also seems to be unnecessarily confusing. Ultimately, specifying optimality requires a combination of algorithm independent lower bounds and as well as upper bounds. The former always necessitates specifying an appropriate class of problem instances. This seems true in general, and for bandits on graphs in particular. Perhaps the point being made here is more subtle, but it remained somewhat elusive to me.
>
> The point is indeed more subtle. Regret is naturally a time horizon-dependent notion. Because we consider finite, time-independent quantities as part of the regret bound, the notion of optimality becomes unclear. Theorem 4.1 and the following discussion give more insight into why optimality is hard to define and what we choose to pursue as optimal regret.
>
> > Overall the structure of the paper is strange. The discussion of related work appears in several places in a somewhat confusing fashion. The proof sketch of Theorem 4.1 appears in Section 6 rather than 4. Section 5 should be probably be moved towards the front of the paper. The two lower bounds could be placed together in a common section.
>
> We thank the reviewer for their comment and will take that into consideration when preparing our final version.

---

### Author Response · Authors · 2022-08-02
**Addressing concerns regarding Theorem 4.1**

There was a typo with $\log(T)$ in the statement of Theorem 4.1 in the denominator, which does not reflect what we prove. The corrected version is: For any $K\geq 32$ and $\Delta_{\min}=\mathcal{O}(\frac{1}{\sqrt{K}})$, there exists a graph with $K$ vertices and an instance with unique best arm and minimal gap $\Delta_{\min}$, such that for any algorithm and any $T\geq \Omega(\frac{K^{3/4}}{\Delta_{\min}})$, the regret is bounded by
\begin{align*}
    \frac{\mathsf{Reg}(T)}{c^*+\frac{1}{\Delta_{\min}}} = \Omega(K^\frac{1}{8}).
\end{align*}
This Theorem implies that for $T = \Theta(\exp(K^\gamma ))$ where $\gamma\in (0,\frac{1}{8})$, there is a polynomial gap between the actual regret and $c^*\log(T)$ of order $K^{\frac{1}{8}-\gamma}$, which means that a regret upper bound of $\mathsf{Reg}(T) = \tilde O(c^\*\log(T)+\frac{1}{\Delta_{\min}})$ is impossible to obtain for this specific family of graphs.
The correction also resolves the contradiction, noted by several reviewers, with respect to the established tight asymptotic lower bound of $c^\*\log(T)$.

---

### Meta-Review · Area_Chair_yiA6 · 2022-08-24

**Recommendation:** Accept
**Confidence:** Certain

**Metareview:**

The reviewers largely agreed in the opinion that the paper has enough results on online learning with feedback graphs. On the other hand, concerns on the presentation and the overall picture of the contribution are raised, which I agree with. Though some of them come from the inherent difficulty of the problem, I strongly encourage the authors to carefully address these points in the final version.

**Award:**

No

---

### Decision · Program_Chairs · 2022-09-14

Accept